# High-performance van der Waals antiferro-electric CuCrP$_2$S$_6$-based memristors

Yinchang Ma [1,6], Yuan Yan [2,6], Linqu Luo [1,6], Sebastian Pazos [1], Chenhui Zhang [1], Xiang Lv[3], Maolin Chen [1], Chen Liu[1], Yizhou Wang[1], Aitian Chen [1], Yan Li [1], Dongxing Zheng [1], Rongyu Lin [1], Hanin Algaidi [1], Minglei Sun [1], Jefferson Zhe Liu [2], Shaobo Tu[1], Husam N. Alshareef [1], Cheng Gong [4], Mario Lanza [1], Fei Xue [5] ✉ & Xixiang Zhang [1] ✉

Layered thio- and seleno-phosphate ferroelectrics, such as CuInP$_2$S$_6$, are promising building blocks for next-generation nonvolatile memory devices. However, because of the low Curie point, the CuInP$_2$S$_6$-based memory devices suffer from poor thermal stability (<42 °C). Here, exploiting the electric field-driven phase transition in the rarely studied antiferroelectric CuCrP$_2$S$_6$ crystals, we develop a nonvolatile memristor showing a sizable resistive-switching ratio of ~1000, high switching endurance up to 20,000 cycles, low cycle-to-cycle variation, and robust thermal stability up to 120 °C. The resistive switching is attributed to the ferroelectric polarization-modulated thermal emission accompanied by the Fowler–Nordheim tunneling across the interfaces. First-principles calculations reveal that the good device performances are associated with the exceptionally strong ferroelectric polarization in CuCrP$_2$S$_6$ crystal. Furthermore, the typical biological synaptic learning rules, such as long-term potentiation/depression and spike amplitude/spike time-dependent plasticity, are also demonstrated. The results highlight the great application potential of van der Waals antiferroelectrics in high-performance synaptic devices for neuromorphic computing.

Ferroelectric materials, with permanent electric dipoles that can be reversed by electric fields, play an important role in advanced information storage devices[1]. Recently, two-dimensional (2D) van der Waals (vdW) ferroelectrics have garnered considerable interest because of their stable polarization, dangling bond-free interfaces, and atomic-scale integration[2,3]. These vdW ferroelectrics have demonstrated rich physics[4,5] and advanced memory performance with brain-like perception–storage–computing functionalities[6]. In particular, thio- and seleno-phosphate ferroelectrics are a class of dielectric materials with wide band gaps. These materials exhibit unique polar physics,

e.g., dipole locking[7], quadruple-potential wells[8] and giant piezoelectric domain walls[9]. Furthermore, they can also be utilized to develop tunnel junctions with high ON/OFF ratios[10–13] and ferroelectric field effect transistors (Fe-FETs) with long retention[14,15].

To date, among the layered thio- and seleno-phosphate ferroelectrics, most previous efforts have been made on CuInP$_2$S$_6$ (CIPS). Since the discovery of ferroelectricity in 4 nm-thick CIPS flakes in 2016[16], a number of architectures for CIPS-based nonvolatile, programmable memory devices have been proposed, delivering high device performance and versatile functionalities toward next-

[1]Physical Science and Engineering Division, King Abdullah University of Science and Technology, Thuwal 23955-6900, Saudi Arabia. [2]Department of Mechanical Engineering, The University of Melbourne, Parkville, Vic 3010, Australia. [3]College of Materials Science and Engineering, Sichuan University, Chengdu 610065, China. [4]Department of Electrical and Computer Engineering and Quantum Technology Center, University of Maryland, College Park, MD 20742, USA. [5]ZJU-Hangzhou Global Scientific and Technological Innovation Center, School of Micro-Nano Electronics, Zhejiang University, Hangzhou 311215, China. [6]These authors contributed equally: Yinchang Ma, Yuan Yan, Linqu Luo. ✉e-mail: xuef@zju.edu.cn; xixiang.zhang@kaust.edu.sa

generation computing hardware. By leveraging CIPS, Wang et al. realized reconfigurable memory devices with long retention time via vdW engineering in a ferroelectric transistor memory cell[17]. Using $MoS_2$/CIPS heterostructures, Si et al. obtained Fe-FETs with stable nonvolatile memory properties[18]. Moreover, by adopting CIPS/graphene heterostructures, gate-tunable memristive effects were discovered[19]. However, because of the low Curie point ($T_c = 42 °C$) of CIPS, high temperatures above 42 °C could destroy the ferroelectric order and induce a transition to the paraelectric order, consequently deteriorating the performance of these memory devices[8,20,21]. This poor thermal stability severely limits the application of thio- and selenophosphate ferroelectric materials in harsh environments.

Despite holding a similar crystal structure to ferroelectric CIPS, $CuCrP_2S_6$ is indeed an antiferroelectric crystal[22,23]; as such, its application in the development of nonvolatile memory devices is challenging because of the lack of remnant polarization[24]. Here, exploiting an electric field-driven phase transition[25–27], we transform the antiferroelectricity of $CuCrP_2S_6$ into ferroelectricity and realize nonvolatile $CuCrP_2S_6$ memristors with two-terminal vertical architectures. Among all reported vdW ferroelectric memristors, $CuCrP_2S_6$ memristors exhibit analog synaptic functionalities, high endurance (up to 20,000 cycles), and high thermal stability (>120 °C). Piezoresponse force microscopy (PFM), second harmonic generation (SHG), and temperature-dependent electrical measurements demonstrate that the resistive-switching behavior is associated with polarization flipping and barrier-height variation. Moreover, first-principles calculations reveal the large polarization and low polarization-reversal energy in $CuCrP_2S_6$, which provides a plausible explanation for the observed low operation voltage, high endurance, and thermal robustness.

## Results and discussion

The crystal structure of $CuCrP_2S_6$ is schematically depicted in Fig. 1a (top view) and d (side view). The metal cations are confined in octahedral frames constructed by sulfur atoms, and the layers are held together by the vdW force. Cu atoms are alternately arranged up and down in the antiferroelectric phase (Fig. 1d, middle panel), whereas in the ferroelectric phase, all Cu atoms are displaced in the up or down position collectively (Fig. 1d, top and bottom panels), resulting in remnant ferroelectric polarization. High-quality plate-like $CuCrP_2S_6$ crystals with typical lateral dimensions of ~5 mm × ~5 mm (inset of Fig. 1b) are synthesized using the chemical vapor transport (CVT) method (see "Experimental" section). The X-ray diffraction (XRD) analysis confirms the high single-crystallinity of the as-grown $CuCrP_2S_6$ crystals (the top panel of Fig. 1b). The Raman spectrum in the bottom panel of Fig. 1b displays phonon peaks at 204, 266, 378, and 591 cm$^{-1}$, agreeing well with the previous report[22]. Energy-dispersive X-ray spectroscopy (EDS) (Supplementary Fig. 1) is applied to confirm the expected elemental composition, with the atomic ratio of Cu, Cr, P, and S approximately 1:1:2:6.

To date, previous studies primarily explored $CuCrP_2S_6$'s low-temperature (<300 K) properties[23,28,29], while its behavior in the high-temperature range remains elusive. Here, employing a variety of characterization techniques including PFM, ferroelectric tester, SHG, and dielectric measurements, we surprisingly discover that the polar states of $CuCrP_2S_6$ can persist at room temperature and above. First, the $CuCrP_2S_6$ flakes are characterized by PFM. As shown in the bottom panel of Fig. 1c, hysteresis loops and "butterfly" curves for the amplitude and phase of piezoresponse are detected for the 10-nm flakes, indicating the alignment of reversible electric dipoles along the out-of-plane direction with remnant polarization, i.e., a signature of ferroelectricity. The top left panel of Fig. 1c shows the PFM phase images after the tip-induced domain reversal in a box-to-box pattern by applying a voltage bias of ±10 V. The clear contrast between the inner and outer parts in the phase image indicates the domain reversal under the PFM tip-induced electric field. Moreover, polarization–electric

field ($P–E$) curves were acquired for a 50-nm thick $CuCrP_2S_6$ flake. Pronounced hysteresis loops (Fig. 1f and Supplementary Figs. 2 and 7) are observed, which unambiguously demonstrate the existence of ferroelectricity. A possible reason for the observed ferroelectricity in this antiferroelectric crystal is that the electric field can induce a structural transition from the antiferroelectric to the ferroelectric phase[25,27,30–32]. Over time, the domains naturally relax back to the ground state, i.e., antiferroelectricity (the top right panel of Fig. 1c).

Second, the ferroelectricity in $CuCrP_2S_6$ crystals was probed using SHG. Figure 1e and Supplementary Fig. 3 show that the poled $CuCrP_2S_6$ flake generates a strong SHG signal, indicating a broken inversion centrosymmetry associated with ferroelectricity. The SHG intensity shows sixfold symmetry, which agrees well with the hexagonal symmetry in the $CuCrP_2S_6$ crystal. $CuCrP_2S_6$ belongs to the space group of P3 (No. 143), consistent with the fact that the SHG intensity can be fitted by $I = I_0 \cos^2(3\theta)$, where $\theta$ is the angle between the incident laser polarization and the a-axis[33]. Notably, some flakes show detectable sixfold SHG intensity even without prior exposure to an electric field, suggesting a possible coexistence of antiferroelectric and ferroelectric coupling.

Supplementary Fig. 4 displays thickness-dependent SHG intensity mapping of $CuCrP_2S_6$ flakes. A thicker sample exhibits a stronger SHG signal, and the decreased SHG responses in thin samples may be attributed to the substrate clamping effect, which is consistent with previous reports of thickness-dependent PFM responses[16]. Besides, Supplementary Fig. 5 shows that SHG intensity is enhanced by electrical poling, revealing the electric-field-induced antiferroelectric-to-ferroelectric transition (details are discussed in Supplementary Note 1). In addition to the SHG results, scanning Kelvin probe microscopy (SKPM) measurement (Supplementary Figs. 6 and 21 and Supplementary Note 2) demonstrates a clear contrast between pristine and poled regions, confirming the ferroelectricity.

The most striking discovery from our material characterization is the observation of a phase transition occurring above room temperature, which was not previously reported. In Fig. 1g,h, a remarkable drop in SHG intensity is observed around 470 K, suggesting a transition from the ferroelectric to the paraelectric phase. In good agreement with this observation, our dielectric measurement shows a clear cusp appearing at 470 K in the dielectric curves (Fig. 1i), further confirming the occurrence of a phase transition around this critical temperature (470 K). Remarkably, previous studies identified two phase transitions at $T_{c1} = 145$ K and $T_{c2} = 195$ K[23,34,35]. Here, we report $T_c = 470$ K as a newly discovered critical temperature for ferroelectric phase transition, above which the dipole coupling completely vanishes. Note that multiple phase transitions are quite common in ferroelectrics when they evolve from ordered to disordered states, resulting in multiple critical temperatures[36]. Therefore, we believe that $T_{c1}$ and $T_{c2}$ are the first two critical temperatures for the dipole coupling strength changes, whereas $T_c = 470$ K may be the actual Curie temperature.

Using a mechanically exfoliated $CuCrP_2S_6$ nanoflake, a vertical memristor crossbar array was fabricated (Fig. 2a–c). The thickness of the nanoflake was 10 nm, characterized by atomic force microscopy (AFM) (Supplementary Fig. 8). Ti and Au were deposited on the top and bottom sides to form asymmetric interfaces (Fig. 2b) for optimizing the resistive-switching behavior[37]. Supplementary Fig. 9 shows the cross-sectional transmission electron microscopy (TEM) image of a typical device. Figure 2d shows the current–voltage ($I–V$) curves for a typical $CuCrP_2S_6$ memristor measured by sweeping voltages between ±1 V. The bipolar resistive-switching behavior is clearly observed with an ON/OFF ratio of ~1000 (read at 0.1 V). Figure 2e shows the expanded hysteretic $I–V$ curves with different sweeping voltages from ±0.1 to ±1 V in a 0.1-V step. Supplementary Figs. 2 and 20 indicate that the required electric field for resistance switching is comparable to that for polarization switching. Figure 2f shows the voltage-sweeping dependence

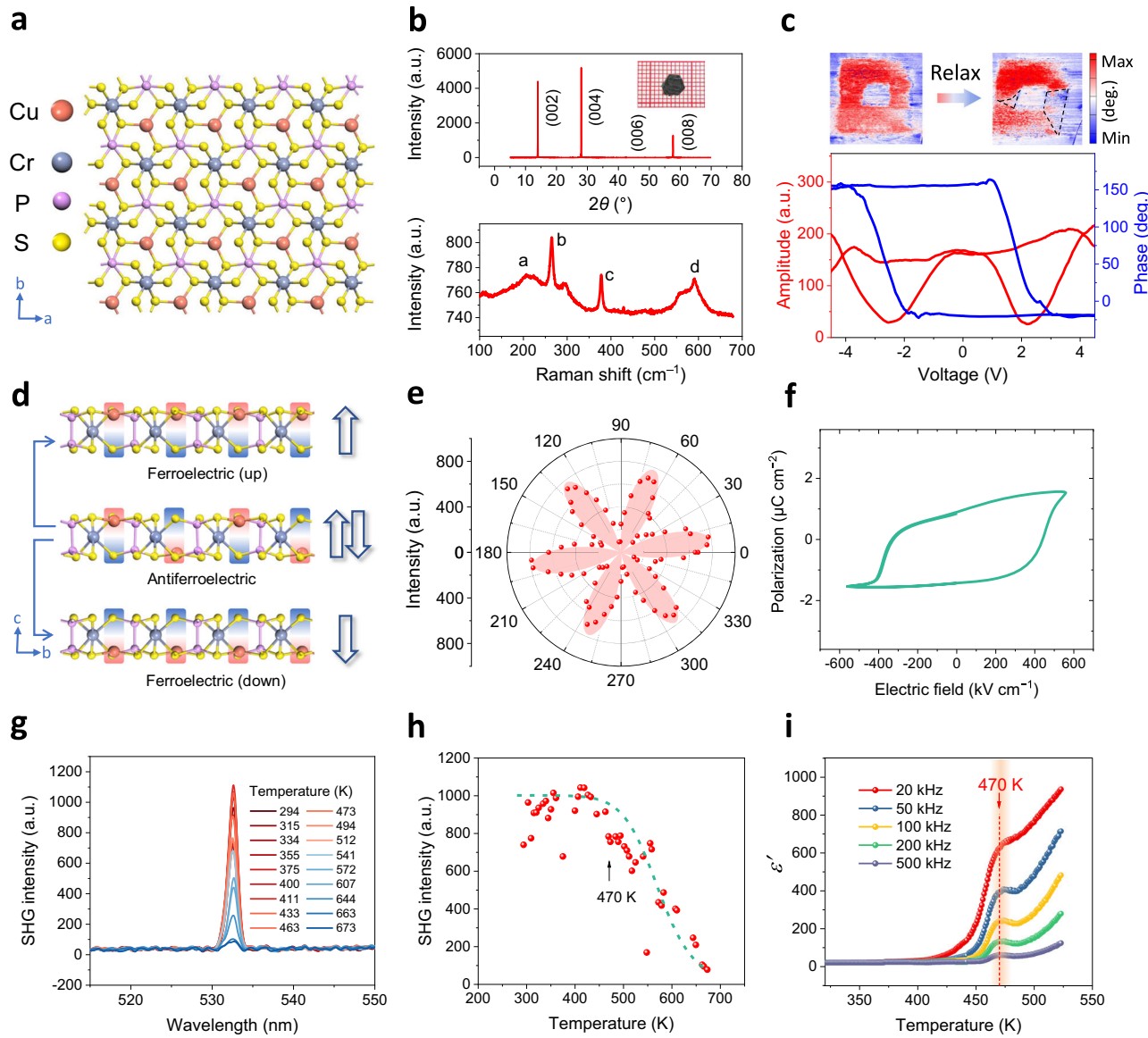

**Fig. 1 | Characterization of CuCrP$_2$S$_6$. a** Crystal structure along [001]. **b** Top panel: X-ray diffraction of CuCrP$_2$S$_6$. Inset: the optical image of the as-grown crystal. Grid size: 1 mm × 1 mm. Bottom panel: Raman spectrum of CuCrP$_2$S$_6$. **c** Top panels: piezoresponse phase image with a written box-in-box pattern before and after the 1-day relaxation. Relaxed parts are marked by black dashed lines. Scan size: 6 μm × 6 μm. Bottom panel: PFM amplitude (red) and phase (blue) hysteresis loops for a 10 nm-thick CuCrP$_2$S$_6$ flake. **d** Cross-sectional crystal structures viewed from *a*-axis in the antiferroelectric phase and ferroelectric phase with polarization directions. **e** Polar plot of SHG intensity for poled ferroelectric CuCrP$_2$S$_6$ flakes. **f** *P*–*E* hysteresis loops measured on a CuCrP$_2$S$_6$ ferroelectric capacitor. Test frequency: 100 Hz. **g** Temperature-dependent SHG intensity measurements. **h** SHG intensity drops at around 470 K. **i** Temperature-dependent dielectric properties of CuCrP$_2$S$_6$ measured at 20, 50, 100, 200, and 500 kHz.

of the currents and the switching ratio read at 0.1 V. The high resistance state (HRS) currents exhibit a weak dependence on the sweeping-voltage variation, whereas the low resistance state (LRS) currents increase by over three orders as the voltage increases from 0.1 to 1 V. Our control experiment shows that the breakdown voltage is around 10 V (Supplementary Fig. 10 and Supplementary Note 3), which is much larger than the voltages we normally applied. Figure 2g shows the stable *I*−*V* curves for 1000 consecutive cycles. Figure 2h, i shows that the HRS and LRS conductance values do not change noticeably after 1000 consecutive cycles, indicating a low cycle-to-cycle variability. Supplementary Fig. 11 shows the LRS and HRS conductance values of all 16 memristors in the 4 × 4 array, which exhibit good device uniformity. To investigate the retention, the as-fabricated memristors are set with pulses of ±1 V/5 s, after which the corresponding resistances are read at a 5-s interval. As shown in Fig. 2j, both HRS and LRS

currents remain nearly unchanged over 5000 s. In addition to retention, endurance is another important figure of merit for memristors. After the DC sweeping measurements, we performed AC pulse measurements. We used AC pulses with an amplitude/width of 2 V/1 ms to test the endurance. The pulse waveform and current variation are shown in Supplementary Fig. 12. Each cycle comprises a positive pulse for "setting" the memristor to the "ON" state and a negative pulse for "resetting" the memristor to the "OFF" state. The "set" or "reset" process is followed by a 1 V/1 ms reading process. Figure 2k, l shows that the LRS and HRS conductance does not degrade even after testing 20,000 cycles. This represents the best endurance performance ever reported among 2D ferroelectric memristors[38,39].

To implement electronic synapses for neuromorphic computing, multilevel conductance states are desirable. We examined the possibility of synapse emulation. A pulse can be used to emulate the

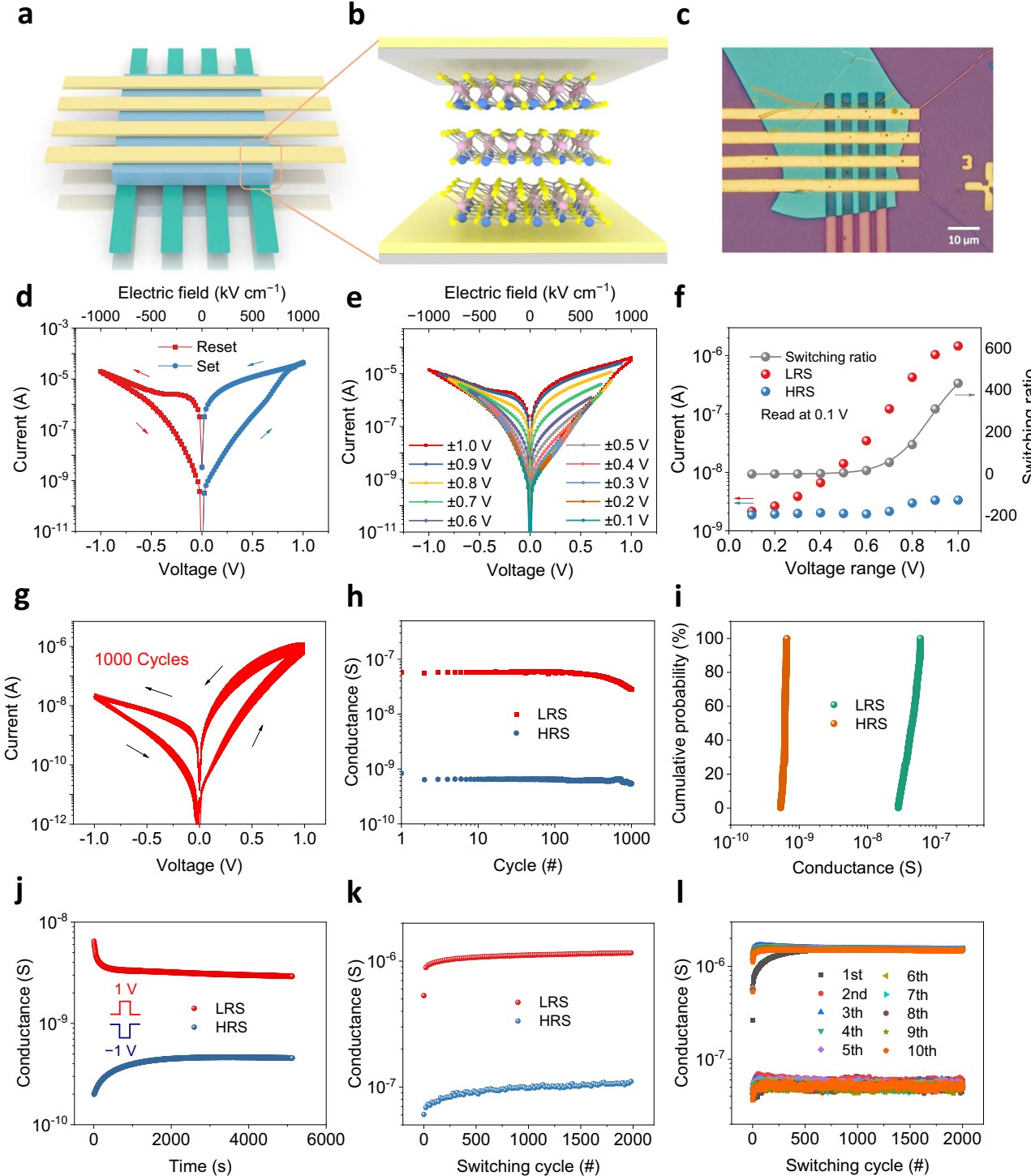

**Fig. 2 | Electrical measurements of the CuCrP$_2$S$_6$ memristor. a** Schematic of the memristor array. **b** Schematic of the CuCrP$_2$S$_6$ memristor. Bottom electrode: Ti/Au (5/15 nm); top electrode: Ti/Au (20/70 nm). **c** Photograph of the memristor array fabricated on a 10-nm CuCrP$_2$S$_6$ flake. **d** Resistive-switching characteristics of the device. **e** $I–V$ hysteresis loops with different sweeping voltages. **f** Evolution of the HRS and LRS currents and switching ratios as a function of the sweeping voltage. The currents are extracted from (**e**) at a voltage of 0.1 V. **g** Endurance test of the resistive-switching behavior by collecting $I–V$ curves for 1000 consecutive sweeping cycles. **h** HRS and LRS conductance over 1000 consecutive cycles. The conductance is read from **g** with a voltage of 0.1 V. **i** Cumulative distribution plot of the HRS and LRS conductance obtained from the 1000 sweeping cycles. The conductance values are from **h**. **j**, Retention time for the HRS and LRS conductance after applying a preset pulse of ±1 V. The reading voltage is 0.1 V, and the reading interval is 5 s. **k** Endurance test for 2000 cycles. The HRS and LRS conductance is measured with the waveform shown in Supplementary Fig. 12. **l** Endurance test for 20,000 cycles (10 tests and each test runs for 2000 cycles).

biological spikes transported between different neurons, and the change of the device conductance can mimic the synaptic weight update between two neurons. In Fig. 3a, the top panel shows the synaptic-weight update (potentiation/depression process) as a function of the continuous pulses, whereas the bottom panel shows the schematics of the applied pulse trains with five cycles. The first cycle comprises 8 identical positive pulses and 8 identical negative pulses, denoted as "8$_{(+)}$ + 8$_{(-)}$" for simplicity. The subsequent four

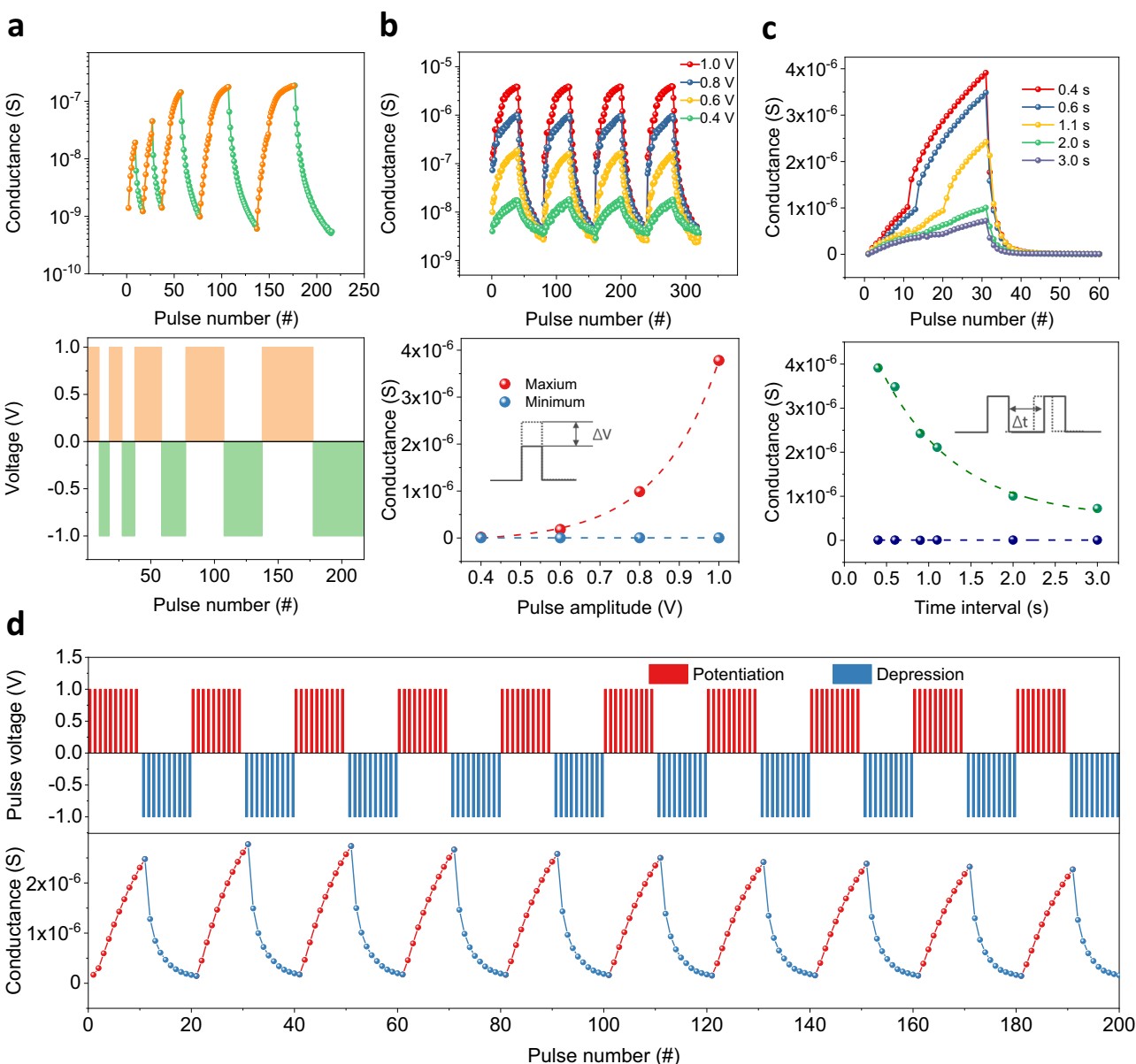

**Fig. 3 | Emulation of the key synaptic behaviors by a CuCrP$_2$S$_6$ memristor.**
**a** Potentiation and depression with different numbers of pulses. The applied voltage pulse sequences are shown in the bottom panel. **b** Emulation of the spike amplitude-dependent plasticity. The pulses with amplitudes of 0.4, 0.6, 0.8, and 1.0 V were applied. The conductance evolution of the device is shown in the top panel. The conductance increases with the pulse amplitude. The conductance maximum and minimum are extracted and plotted in the bottom panel. A large pulse amplitude enables the conductance to reach a higher level. **c** Spike rate-dependent plasticity. The pulses are set to be 1 V/300 ms, and the intervals between pulses are set as 0.4, 0.6, 1.1, 2.0, and 3.0 s. The pulse-rate dependence of the conductance maximum and minimum are extracted and plotted in the bottom panel. Insets of **b** and **c** show the variables (pulse–amplitude and pulse–time interval) in the measurements. **d** Cycling test for the potentiation/depression behaviors. Set/reset voltage: ±1 V; read voltage: 0.1 V.

cycles are denoted as $10_{(+)} + 10_{(-)}$, $20_{(+)} + 20_{(-)}$, $30_{(+)} + 30_{(-)}$, and $40_{(+)} + 40_{(-)}$, respectively. Each pulse is followed by a voltage of 0.1 V to read the device's conductance. As shown in the top panel of Fig. 3a, the conductance steadily increases under positive-pulse excitation, thus emulating the "potentiation" process of the synapses; however, it gradually decreases under negative-pulse excitation, thus emulating the "depression" process. An increasing number of pulses produce additional intermediate resistance states, indicating the effective modulation of the device conductance through electrical pulses. These results show that consecutive pulse sequences unambiguously exert a training effect on the device conductance, which is referred to as the "synaptic behavior".

We continued to investigate the dependence of conductance evolution on the applied voltage amplitude and interval. Figure 3b

shows the evolution of the device conductance stimulated by pulse sequences with different voltage amplitudes of 0.4, 0.6, 0.8, and 1 V. The conductance is gradually potentiated by the positive voltage pulses, whereas it is gradually depressed by the negative voltage pulses. Evidently, a large pulse amplitude corresponds to a large conductance. The conductance values reach 0.017, 0.18, 1.0, and 3.9 μS with pulse amplitudes of 0.4, 0.6, 0.8, and 1.0 V, respectively. The minimum/maximum conductance versus the pulse amplitude is plotted in the bottom panel of Fig. 3b. With increasing pulse (spike) amplitude, the maximum conductance exponentially increases, whereas the minimum conductance remains almost unchanged. Furthermore, as shown in Fig. 3c, with a relatively high spike rate (short time interval between pulses), the conductance is potentiated to a high level. Unlike the spike-amplitude dependence shown in Fig. 3b, the

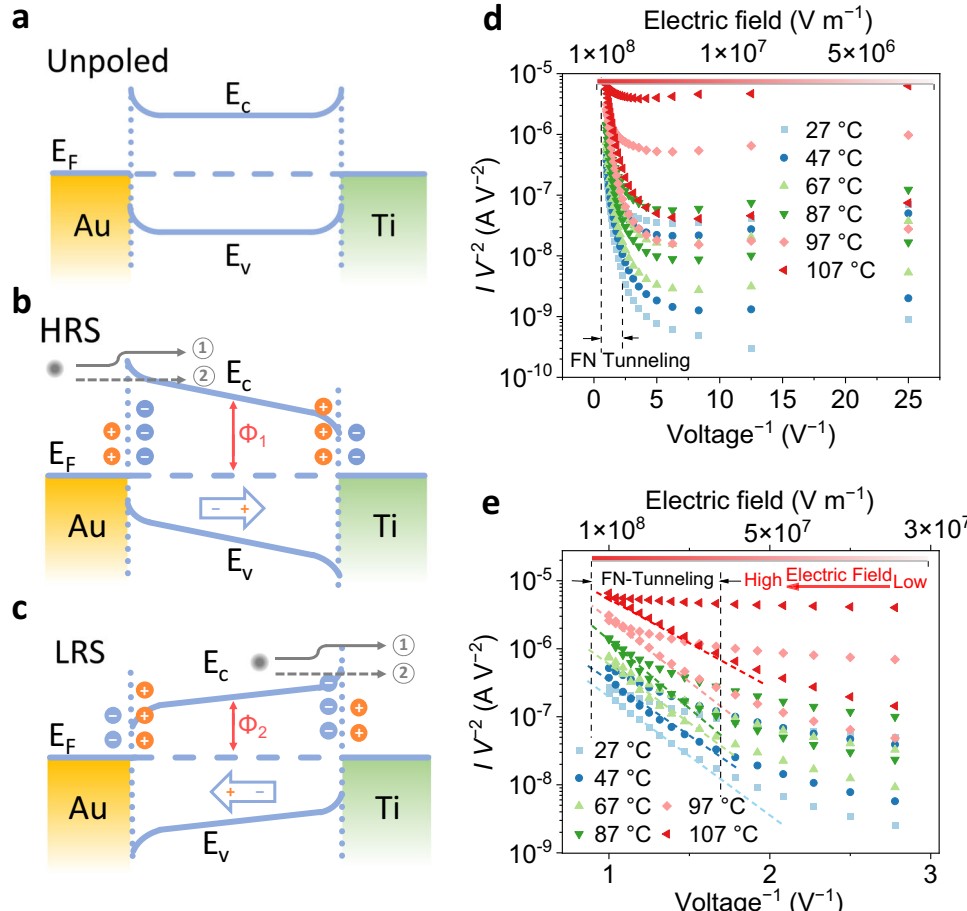

**Fig. 4 | Schematics of the mechanisms of resistive switching and the temperature-dependent electrical transport measurements. a–c** Energy-band diagrams of the CuCrP$_2$S$_6$ memristor in the unpoled state, HRS, and LRS, respectively. The energy band tilts in different directions, corresponding to different polarization states. Two conduction mechanisms (① Schottky emission and ② FN tunneling) are marked by arrows in (**b**) and (**c**). The average energy barriers are marked as $\Phi_1$ and $\Phi_2$ ($\Phi_1 > \Phi_2$), which lead to HRS and LRS, respectively. **d** Temperature-dependent electrical-transport measurements plotted in the form of $I/V^{-2}$ vs. $V^{-1}$. The region marked by the dashed line indicates the occurrence of FN tunneling at high voltages. **e** Fitting of the measurement results in (**d**) with the FN-tunneling model. $\ln(I/V^{-2})$ and $V^{-1}$ follow a linear relation in the marked region, indicating the occurrence of the FN tunneling within the range of 0.45–1 V.

bottom panel of Fig. 3c highlights that the maximum conductance exponentially decreases as the pulse interval increases (corresponding to low spike rates), whereas the minimum conductance only undergoes a slight change. To summarize, the conductance of the memristor can be effectively potentiated and depressed by modulating the pulse intervals, amplitudes, and numbers, which emulates the synaptic weight update under stimuli from pre-neuron and post-neuron spikes. Based on the observed potentiation and depression behaviors, we examine the repeatability of the training effect. The pulse sequence comprising ten "$10_{(+)} + 10_{(-)}$" cycles is shown in Fig. 3d, where the potentiation and depression processes are marked in red and blue, respectively. All intermediate states are distinct, uniform, and reproducible for all ten cycles, demonstrating the good repeatability of the training effect.

The flipping of the ferroelectric polarization is responsible for the observed resistive switching. The resistive-switching mechanism can be interpreted by using the asymmetric energy-band model[40]. Figure 4a–c shows the energy band diagram of the device in the unpoled state, HRS, and LRS, respectively. An electric field transforms CuCrP$_2$S$_6$ from an antiferroelectric state into a ferroelectric state and enables the switching of the ferroelectric dipoles. Furthermore, it modifies the band alignment at the interfaces by tilting the energy band of CuCrP$_2$S$_6$, which consequently modulates the energy barrier involved in the electron-conduction process[37]. Note that, due to the

ionic nature of CuCrP$_2$S$_6$, ferroelectric polarization switching may be accompanied by ion migration and ion migration-related charge trapping/detrapping effects during the resistive switching, which is often observed in oxide ferroelectric memory devices[41,42]. However, the ion-related filamentary switching is firmly ruled out by our control experiments (Supplementary Fig. 13 and Supplementary Note 4).

To understand the mechanism of the electron conduction, temperature-dependent electrical transport measurements were performed from room temperature to 107 °C (380 K). In previous studies of 2D ferroelectric memristors, the Schottky emission was considered the main conduction mechanism[43]. However, Supplementary Fig. 14 shows that the Schottky model cannot explain our results in the strong electric field region. Therefore, the electrical transport is unlikely to be governed by the thermal excitation of carriers, particularly in the strong electric field (the detailed analyses are presented in Supplementary Note 5 and Supplementary Fig. 15). We collected a set of $I$–$V$ curves (Supplementary Fig. 16) for samples with varied thicknesses. The resistive switching ratio is enhanced as the sample thickness decreases. Intuitively, a decrease in thickness down to the nanoscale brings the sample closer to the critical thickness of ferroelectricity, and thus a larger interfacial depolarization field can deteriorate the ferroelectric behavior and related resistive switching. Therefore, an improvement in device performance with reduced thicknesses implies that the origin of resistive switching is not only based on interface

effects. Considering that the $CuCrP_2S_6$ flakes with low thicknesses (~10 nm) could allow electron tunneling, we expect that both the Schottky emission and Fowler–Nordheim (FN) tunneling will contribute to the electron conduction in this device[44]. The FN tunneling current can be expressed as follows:

$$I \propto V^2 \exp\left(-\frac{4\sqrt{2m^*}(q\Phi)^{\frac{3}{2}}}{3q\hbar V}\right), \qquad (1)$$

$$\ln\left(\frac{I}{V^2}\right) \propto \left(-\frac{1}{V}\right)\left(\frac{4\sqrt{2m^*}(q\Phi)^{\frac{3}{2}}}{3q\hbar}\right), \qquad (2)$$

where $I$ is current, $\hbar$ is reduced Planck's constant, $m^*$ is effective mass, $V$ is applied voltage, and $\Phi$ is barrier height. The current was measured at different temperatures by sweeping the voltage in the range of ±1 V, and the data are plotted in the form of $I\,V^{-2}$ vs. $V^{-1}$ in a logarithmic scale (Fig. 4d, e). The apparent linear dependence between $\ln(I\,V^{-2})$ and $V^{-1}$ in the strong electric-field region (details are shown in Supplementary Note 6) indicates that the FN tunneling dominates the electrical transport mechanism under the strong electric field as expected. The physical processes of the FN tunneling are illustrated in Fig. 4b, c, marked as ②, in contrast with the Schottky-emission processes marked as ①. Under a strong electric field, both the height and width of the tunneling barrier reduce, enhancing the electron tunneling transmission (Supplementary Fig. 17), which is consistent with the fact that the FN tunneling results from the barrier-narrowing effect. These results confirm that tunneling conduction indeed contributes to resistive switching beyond the well-recognized ordinary thermal emission[4].

Having identified the physics behind the nonvolatile memory effect, we still expect to unveil the mechanism underlying the synaptic behaviors (potentiation/depression process). Because all the above-mentioned observations are related to the ferroelectric polarization switching in $CuCrP_2S_6$, we deduce that the potentiation and depression processes are correlated with the ferroelectric domain dynamics[21,45]. To experimentally confirm this postulation, we transfer $CuCrP_2S_6$ flakes onto the bottom electrode connected to a pulse generator and subsequently performed PFM measurements to map out the domain-evolution processes (Fig. 5e). Figure 5a–d shows the piezoresponse amplitude images under consecutive pulses. The flake was initially poled with a long pulse of −8 V/10 s, thus resulting in a uniform dark image. This indicates that the entire ferroelectric domains were completely reversed downward, thus forming a single domain state (Fig. 5a). With increasing pulse (8 V/0.5 s) numbers, the upward domain was first nucleated at the left and subsequently propagated to the right (Fig. 5b, c), during which an increasing number of dipoles were aligned upward (along the electric-field direction). Finally, after >20 pulses, the entire sample was fully poled upward through the domain-wall motion. The sample became a single-domain state (Fig. 5d) but with a polarization opposite to that of the initial state. The process of increasing the upward domains with the number of positive pulses should correspond to the potentiation process. On the contrary, if we continuously apply negative pulses, the size of the down domains will increase, corresponding to the depression process. Figure 5a, d corresponds to the HRS and LRS, respectively, while Fig. 5b, c corresponds to the intermediate transition states. Therefore, many stable states indeed exist between the LRS and HRS. Based on these observations, macroscopic synaptic characteristics can be correlated with the microscopic dynamics of these domains[46].

To examine the thermal stability of these memristors, we tested their performances at high temperatures. Figure 6a shows that the switching window remains open from room temperature to 120 °C, demonstrating the excellent thermal stability of the memristor. Furthermore, this excellent stability satisfies the requirements for most practical applications, because the temperature of common working

environments for microprocessors is ~80 °C[47]. As the temperature increases further, the resistive-switching ratio gradually decreases and finally vanishes at 150 °C (Supplementary Fig. 18), which is consistent with the temperature-dependent Raman measurements (Supplementary Fig. 19). The absence of resistive-switching behavior at high temperatures conforms to the nature of ferroelectric–paraelectric transition in ferroelectrics. Using first-principles calculations, we attribute the superior performance of our memristors to the inherent polar nature of $CuCrP_2S_6$. After optimizing the crystal-structure model of $CuCrP_2S_6$ and CIPS (Fig. 6b) using density functional theory (DFT) calculations, we calculated their ferroelectric polarization strengths (Fig. 6c) for one unit cell with one formula unit (f.u.). Remarkably, $CuCrP_2S_6$ possesses a much stronger polarization ($3.4757 \times 10^{-11}$ C m$^{-1}$) than its counterpart, CIPS ($4.5168 \times 10^{-12}$ C m$^{-1}$), which explains its robust performance in the device. Furthermore, we calculated and compared the energy barriers for the polarization-reversal processes in $CuCrP_2S_6$ and CIPS with the nudged elastic band (NEB) method. The energy barriers are 0.1894 eV and 0.3128 eV per f.u. for $CuCrP_2S_6$ and CIPS, respectively (Fig. 6d). This explains the low-voltage operation of the $CuCrP_2S_6$ memristor. Based on these calculation results, we ensure that the electric dipoles are strongly stabilized and coupled in the $CuCrP_2S_6$ memristors, which successfully explains their superior stability.

In summary, we demonstrated an approach to electric-field driven phase transition for implementing antiferroelectric vdW crystals in nonvolatile memory devices using $CuCrP_2S_6$. We demonstrated the multiple distinctive features of the $CuCrP_2S_6$ memristor, including a high ON/OFF ratio (~10³), excellent endurance (20,000 cycles), high thermal stability (>120 °C), a low operation voltage (1 V), and the ability to mimic synaptic behaviors. We investigated the polarization reversal, domain evolution, and electrical transport mechanisms in the memristors, which provides a fundamental understanding for improving antiferroelectrics-based resistive-switching devices. Our work expands the library of 2D materials for nonvolatile memory and synaptic devices, which opens an avenue to high-performance memristors and brain-inspired in-memory computing hardware.

## Methods

### Material growth and characterization
High-quality single crystals of $CuCrP_2S_6$ were synthesized by the CVT method. Cu (99.99%, Sigma-Aldrich), Cr (99.99%, Sigma-Aldrich), P (99.999%, Sigma-Aldrich), and S (99.98%, Alfa Aesar) in the stoichiometric proportion of 1:2:3:6, with a total mass of 1 g, were mixed and employed as precursors. To achieve the transport of matter, 80 mg of iodine serving as the transport agent was added to the precursors. The mixture was sealed in a quartz ampule (inner diameter: 10 mm) inside a glove box filled with Ar. Then, the quartz ampule was tightly connected to a mechanical pump and pumped to $10^{-4}$ mbar before being sealed again. Subsequently, the ampule was transferred to a furnace with two temperature zones. The hot and cold ends were set to 750 °C and 700 °C, respectively, establishing a temperature gradient of 3 °C/cm. The thermal treatment lasted for seven days, at the end of which the furnace was switched off and allowed to cool to room temperature at a very low rate of 0.7 °C/min. The single-crystal XRD pattern was recorded using a Bruker D8 Advance X-ray diffractometer with CuKα radiation. Raman measurement was performed using a 633-nm excitation laser on a WITec alpha300 apyron confocal Raman microscope. The laser power was maintained below 0.5 mW to avoid local laser-induced heating.

### Device fabrication
Thin-bottom electrodes were patterned by electron-beam lithography (EBL: CABL-9000C), followed by metal sputtering (5 nm Ti/10 nm Au) and lift off on a 300 nm $SiO_2$/Si substrate. The $CuCrP_2S_6$ crystal was exfoliated on a Si wafer and subsequently transferred to the bottom

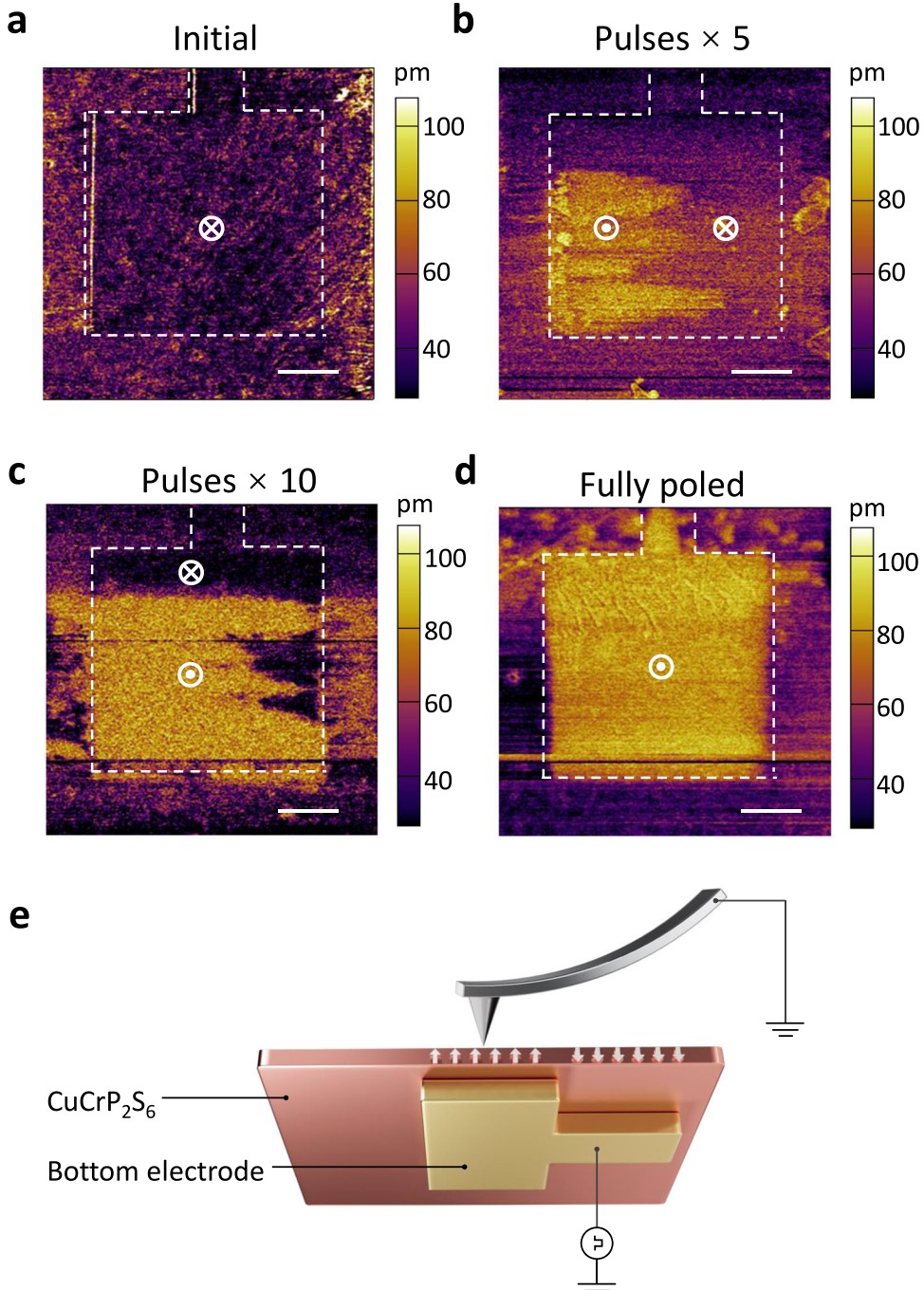

**Fig. 5 | Observation of the intermediate transition states using PFM.**
**a**–**d** Piezoresponse-amplitude image of the domain-evolution processes. Scale bar: 1 μm. **a**, **d** The HRS and LRS of the memristor, respectively. **b**, **c** Intermediate resistance states. The outlines of the bottom electrode are marked by the white dashed lines. **e** Experimental setup for the PFM measurements. The PFM tip is grounded, and the bottom electrode is connected to a pulse generator.

electrode using the standard dry-transfer technique with a polymer stamp. Subsequently, the top electrodes (20 nm Ti/70 nm Au) were deposited by e-beam evaporation after the EBL patterning.

**Ferroelectric characterization**
PFM measurements were performed using a scanning probe microscope (Asylum Research MFP-3D) in the dual-AC resonance mode on conductive gold-coated substrates. A conductive tip (coated with Pt/Ir, spring constant: 3 N m$^{-1}$, produced by Bruker) was employed. The PFM mapping was acquired with a 0.8 V AC bias applied to the probe. SHG measurements were performed using an MStarter 100 Ultrafast SHG microscope spectrometer (Nanjing Metatest Optoelectronics corporation) with an excitation light source of 1064 nm picosecond pulse laser. The ferroelectric hysteresis (*P–E*) loops were measured using a commercial ferroelectric workstation (Radiant Technologies, Inc. Precision Premier 200 V ferroelectric test system), and the measurement frequency was set as 0.1 Hz.

**Device characterization**
All electrical measurements of the memristor were performed using both Keithley 4200 and Agilent B1500A semiconductor parameter analyzers. The *I–V* curves were collected in a quiet sweep mode using Keithley 4200. All electrical measurements were performed in air at room temperature.

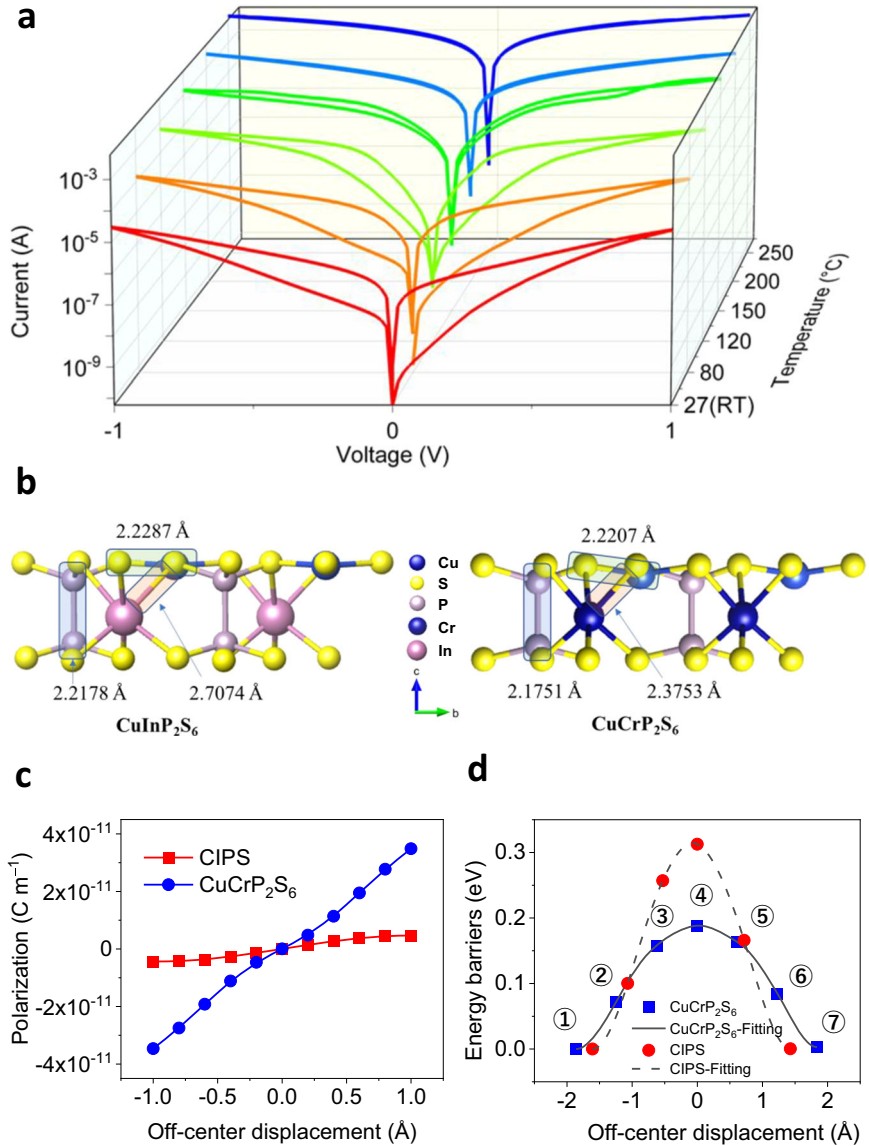

**Fig. 6 | Temperature-dependent *I–V* characteristics and DFT calculations of CuCrP$_2$S$_6$ and CIPS. a** *I–V* characteristics of the device recorded from room temperature (RT) to 250 °C. **b** DFT-optimized crystal structures of CIPS and CuCrP$_2$S$_6$ viewed from *a*-axis in the ferroelectric phase. The bond lengths are marked. **c** DFT-calculation results for the ferroelectric polarization of CuCrP$_2$S$_6$ and CIPS. **d** DFT-calculation results for the polarization-reversal energy barriers in CuCrP$_2$S$_6$ and CIPS.

## First-principles calculations

First-principles calculations were performed using the Vienna ab initio Simulation Package. We adopted the projector-augmented wave method[48] and the Perdew–Burke–Ernzerhof exchange-correlation functional[49]. The cut-off of the plane-wave kinetic energy was set to 400 eV. For all structural relaxations, $k > \frac{25}{a} \times \frac{25}{b} \times 1$ gamma-centered k-point grid was used to sample the Brillouin zone, where $a$ and $b$ are the lattice constants of the supercell. Then, a precise k-point mesh of $k > \frac{60}{a} \times \frac{60}{b} \times 1$ was used for all static calculations. A large vacuum space of ≥20 Å in the direction of $c$ was applied to avoid any spurious interaction between the periodically repeating layers. All structures were completely relaxed until the energy converged within $10^{-6}$ eV and the forces within $10^{-3}$ eV Å$^{-1}$. The climbing-image nudged elastic band method[50] was employed to calculate the phase-transition barriers.

## Data availability

All the data used to reach the conclusion of this study are presented in the paper and the Supplementary Materials. Source data are available from the Figshare repository under https://doi.org/10.6084/m9.figshare.24459703.

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

## Acknowledgements

This publication is based on research supported by the King Abdullah University of Science and Technology, Office of Sponsored Research (OSR), under award numbers: OSR-2017-CRG6-3427, OSR-2019-CRG8-4081 and "Semiconductor Initiative". F.X. thanks for the support from the ZJU-Hangzhou Global Scientific and Technological Innovation Center with startup funding (02170000-K02013012).

## Author contributions

Y.M., Y.Y., and L.L. contributed equally to this work. Y.M. and F.X. conceived and coordinated the project. Y.M. and M.L.C. fabricated the devices. Y.M., L.L., and S.P. performed the electric measurements and analysis. Y.M. and C.L. performed the STEM characterization. Y.M. and X.L. carried out the PFM measurements. S.T. and H.N.A. conducted ferroelectric measurements on a ferroelectric tester. Y.M., F.X., Y.W., R.L., H.A., C.Z., A.C., Y.L., D.Z., and M.L. contributed to the analysis of the experimental data. Y.M., Y.Y., M.S., and J.Z.L. performed DFT simulations. C.G. provided suggestions on writing. The study was supervised by F.X. and X.Z. All authors contributed to the discussion and revision of the paper.

## Competing interests

The authors declare no competing interests.
