## [Peer Review File · Nature Communications]

REVIEWER COMMENTS

Reviewer #1 (Remarks to the Author):

This paper is potentially interesting but a lot of clarification is needed. The system under discussion is an antiferroelectric and as such should not exhibit any macroscopic piezoresponse or spontaneous polarization. Application of an electric field (which in this case is HUGE, 10V/10nm) is used to push it into the ferroelectric state. This is not by itself new, since all such AFE systems exhibit a field induced transition into the FE state. BUT, it is critical to demonstrate how stable this field induced FE state is. It should be relaxing into the AFE state with time. This data needs to be presented. I am also concerned about the magnitude of the applied field: One wonders whether the crystal would undergo a breakdown. From an experimental perspective, I would like to have seen more than just a PFM data, mainly because the PFM technique is prone to several sources of spurious signals. Thus, establishing the FE state with SHG or some other technique that probes the FE state would be crucial. Without this, the data is interesting but not complete.

Reviewer #2 (Remarks to the Author):

The authors are pursuing resistive switching functionality in the emerging class of transition metal thiophosphates. This family of materials has yet to deserve the same level of attention as structurally simpler TMDs, and finding new functionalities is a great way to motivate interest in the topic. However, the authors are attempting to connect the observed resistive switching in CuCrP2S6 to supposed antiferroelectric activity. And it is in this respect that the paper falls short. In my opinion, the presented data cannot be considered as evidence for antiferroelectricity. Meanwhile, resistive switching is reasonably robust. If the authors can find a way to explain RS without polar transitions, or support the existence of transitions in a more definitive way, the paper can be reconsidered for the publication.

Specifically:

(1) The simplest argument why polarization activity cannot be the reason for observed RS properties is that the transition temperatures for CuCrP2S6 are in the sub 200K range. A very nice recent study (<https://onlinelibrary.wiley.com/doi/full/10.1002/adfm.202204214>) of this material clearly shows this. Also earlier reports on CuCrP2S6 report low transition temperature

(<https://doi.org/10.1016/j.ssc.2005.05.040>). Meanwhile, present studies were carried out at room temperature and above.

(2) Scanning probe microscopy such as PFM is a notoriously poor technique to detect polarization and ferroelectric properties in materials without well-defined domain structure (<https://aip.scitation.org/doi/10.1063/1.4979015>). This is certainly the case in present study. The images in Figure 5 reveal obscure boundaries of poled regions, that are also "streaky" in the direction of the tip scanning. There are ways to extend PFM, using a combination with KPFM. However, to prove polarized structures, the best measurement would be second harmonic generation. Both on pristine and modified areas.

(3) The authors never consider ionic conductivity as the origin or RS. However it is most likely what's causing the observed resistive effects. Generally I expect that any field-induced processes reported in this paper will strongly change the chemical structure (e.g. topography in Fig. 5). The authors should either provide robust evidence to the contrary, or embrace structural change as the source of RS and thoroughly revise the hypothesis and explanation of their work.

Reviewer #3 (Remarks to the Author):

Ma and coworkers report on the electric field induced ferroelectric phase transition in $\text{CuCr}_2\text{P}_2\text{S}_6$ (CCPS) and the integration of such van der Waals materials in a capacitor design for memristive type application. The higher ferroelectric transition temperature, compared to existing van der Waals systems allows for broader temperature range for applications. The authors demonstrate a high ON/OFF ratio in the resistance change controlled by the ferroelectric polarization reversal.

Controlling the ferroelectric domain population allows for beyond binary response and hence for neuromorphic type applications.

I find the manuscript interesting, the push for energy efficient electronic devices is in my opinion of high interest. Here, in particular I appreciate the direct demonstration of the controlled resistance states driven by the domain population, directly imaged by scanning probe microscopy.

I would recommend this work for further consideration once the authors address the following points:

- Several sections of the manuscript may appear confusing to non experts. In the introduction, the “Cu atoms in disorder” may need further disorder and context.
- The poor thermal stability may not be correlated to the ferroelectric transition temperature. Could the authors be more specific? Do they address the chemical stability (e. g. degradation of the structure) at elevated temperature or, specifically the increase of ferroelectric T_c ?
- The term of “giant” polarization is not suitable here. Reports on giant polarization refer to values exceeding 150 $\mu\text{C}/\text{cm}^2$. (Phys. Rev. Lett. 107, 147602 (2011)).
- Was the polarization of CCPS measured? Using PUND? Could the authors show a P-E loop?
- The authors attribute the change of resistance to the reversal of polarization and use the Schottky emission and FN tunneling as main mechanism for the transport.
- Can't the authors consider tunneling electroresistance? Here the asymmetric tunneling potential and tunneling electroresistance may explain as well the polarization dependent resistance change?

I would recommend the authors to refer to the studies Nature 460, 81 (2009), Science 313, 181–183 (2006) among many others.

- Here, controlled domain population also leads to several levels of resistance and studies Nature Commun. 13, 3159 (2022) may deserve some reference too in the context of the work presented.
- Ideally a thickness dependence of the ferroelectric material in transport properties may help identifying the mechanism. Have the authors tried different thicknesses? How do the I/V curves evolve as the films get thicker?
- The PFM data in 1f seem to show 3 contrast levels, which do not appear later in figure 5.

Could the authors show the time dependence of the phase or alternatively try to optimize the AC reading bias to get sharper contrast?

Response Letter

Dear all reviewers,

Thank you for the constructive and insightful comments on our manuscript entitled "Van der Waals Antiferroelectric CuCrP₂S₆-based Memristors with Low Operation Voltage, High Endurance, and High Thermal Stability" (No. NCOMMS-23-00989). We have carefully considered, discussed, and addressed each point raised by the reviewers, and then accordingly revised our manuscript. The replies and revisions are presented as follows and highlighted in the revised manuscript.

With these revisions, our manuscript has been significantly improved.

With best regards,

Fei Xue, PhD

ZJU-Hangzhou Global Scientific and Technological Innovation Center

School of Micro-Nano Electronics

Zhejiang University, Hangzhou 311200, China.

Email: xuef@zju.edu.cn

Xixiang Zhang, PhD

Physical Science and Engineering Division

King Abdullah University of Science and Technology

Email: xixiang.zhang@kaust.edu.sa

Response to Reviewer #1

This paper is potentially interesting but a lot of clarification is needed. The system under discussion is an antiferroelectric and as such should not exhibit any macroscopic piezoresponse or spontaneous polarization. Application of an electric field (which in this case is HUGE, 10V/10nm) is used to push it into the ferroelectric state. This is not by itself new, since all such AFE systems exhibit a field induced transition into the FE state. BUT, it is critical to demonstrate how stable this field induced FE state is. It should be relaxing into the AFE state with time. This data needs to be presented. I am also concerned about the magnitude of the applied field: One wonders whether the crystal would undergo a breakdown. From an experimental perspective, I would like to have seen more than just a PFM data, mainly because the PFM technique is prone to several sources of spurious signals. Thus, establishing the FE state with SHG or some other technique that probes the FE state would be crucial. Without this, the data is interesting but not complete.

Thank you for the constructive suggestions. We fully agree that it is critical to demonstrate the stability of FE state, and provide additional characterizations, such as SHG. We believe that the following response gives a sufficient explanation for the comments.

Evidence for the FE stability (nonvolatile AFE–FE transition)

1. **PFM measurement confirms the durability of ferroelectricity.** By writing a box-in-box pattern with the voltage applied onto a CuCrP_2S_6 flake via the PFM tip (Figure R1a), a remarkable contrast is observed, as previously reported in ferroelectric materials. Moreover, even after 24 hours, the pattern still persists (Figure R1b). This behavior can rule out artifacts such as the electrochemical effect, and indicates that, in our experiments, the stable FE phase has been acquired.

Figure R1. PFM phase images measured with 5-minute (a) and 24-hour (b) delays after writing a

box-in-box pattern.

2. **The measurement using a commercial ferroelectric tester (aixACCT TF Analyzer 2000, Germany) signifies stable ferroelectricity, rather than antiferroelectricity.** We fabricated a ferroelectric capacitor by patterning electrodes using electron beam lithography (EBL) on a 50-nm thick CuCrP_2S_6 flake. After poling the flake with a DC voltage of 2 V, we measured polarization–voltage (P–V) curves with 10-minute and 1-hour delays, respectively. Pronounced hysteresis loops (**Figure R2**) are observed, which unambiguously demonstrates the existence of stable ferroelectricity.

Figure R2. P–V hysteresis loops measured by ferroelectricity tester on CuCrP_2S_6 capacitor with 10-minute (a) and 1-hour (b) delays after electrical poling. Test frequency: 0.1 Hz.

3. **First-principles calculations reveal the possibility of creating stable FE in CuCrP_2S_6 .** From the energy pathway (**Figure R3**), we observed that an energy barrier of 50 meV hinders the relaxation from FE to AFE state. The lowest energy path of the AFE–FE transition calculated for one unit cell with the nudged elastic band (NEB) method is shown in Figure R3b. The energy of FE state is 0.18 eV (Φ_1) higher than that of the AFE ground state, suggesting that electric fields can overcome this barrier and prompt the AFE–FE transition. Noticeably, in the transition processes, many intermediate states and two energy barriers (Φ_2 : 0.05 eV and Φ_3 : 0.04 eV) appear, preventing the FE from relaxing back to AFE. Moreover, the crystal structures corresponding to the six states in Figure R3b are schematically shown in Figure R3a.

Figure R3. AFE–FE transition dynamics and its energy pathway. (a) Side view of CuCrP_2S_6 crystal structures undergoing AFE–FE transition upon applying an upward electric field. (b) DFT calculations of lowest energy pathway for AFE–FE transition process of CuCrP_2S_6 with the NEB method. State ① and state ⑥ represent AFE and FE states, respectively. Φ_1 , Φ_2 , and Φ_3 are 0.18 eV, 0.05 eV, and 0.04 eV, respectively.

To verify the ground state at finite temperatures, we performed DFT calculations on the free energy while taking entropy into account. The Gibbs free energy curves at various temperatures are plotted in **Figure R4**. The AFE state remains energetically favored over the FE state in all investigated temperatures, and their energy difference is small and decreases with increasing temperature, making the transition from AFE to FE more favorable.

Figure R4. Calculated Gibbs free energy for AFE and FE phases of CuCrP_2S_6 .

To theoretically evaluate the FE stability, we performed a phonon dispersion analysis for both AFE and FE states. The phonon spectra are shown in **Figure R5**, indicating the absence of imaginary frequencies. Both AFE and FE states exhibit dynamic stability. The calculations were conducted without an external electric field, suggesting that the FE state can maintain its polarization even after removing the electric field.

Figure R5. Calculated phonon spectra for (a) AFE and (b) FE phases of CuCrP_2S_6 .

Figure R6. Differential charge density of FE- CuCrP_2S_6 . The red and green bubbles represent charge accumulation and depletion, respectively.

A great number of screening charges within the electrodes could contribute to stabilizing ferroelectric dipoles [*Science* 353, 274-278 (2016)], **making it unlikely to return to AFE**. From differential charge density calculated by DFT (**Figure R6**), we observe electron redistribution. Off-centered Cu atoms lose electrons to form positive charge centers. Simultaneously, asymmetric Cu atoms rearrange the electron cloud of the whole system. It is worth noting that the electrons donated by Cu are not captured by the adjacent S atoms, and instead, they transfer to the lowest S atoms via the middle Cr atoms. Strong covalent bonds form between the underlying S atoms and their neighboring Cr atoms. The bond lengths between Cr and S atoms in the lower part are 2.33261 and 2.35040 Å, respectively, which are smaller than the bond length between S atoms and Cr atoms in the upper part (2.45108 and 2.46931 Å). This Cr-mediated electron cloud redistribution can increase the offset of positive and negative charge centers, which is an important reason for the high polarization in CuCrP₂S₆. These electron cloud redistribution and covalent bond forming processes are almost impossible to be reversible without external perturbation.

4. The FE states induced by external fields can be stable even after withdrawing them, which has been recently reported in antiferroelectric oxides [*Appl. Phys. Lett.* **118**, 132903 (2021)]. In this literature, three stages (AFE P phase → FE Q phase without macroscopic polarization and strain → FE Q phase with macroscopic polarization and strain) during the AFE–FE transition were observed, and the transformation dynamics are completely decoupled. Similar AFE–FE transition with stable FE states was also reported in [*Acta Materialia* **200**, 127-135 (2020)], in which the ground state is AFE and the material exhibited stable characteristic FE switching loops after poling, indicating an irreversible AFE–FE transition. Besides these works, there are other studies demonstrating the irreversible switching of AFE-to-FE in both bulk and 2D materials, for example,

- 1) (Pb_{1-x}La_x)(Zr_{0.90}Ti_{0.10})_{1-x/4}O₃ ceramics:
[*J. Eur. Ceram. Soc.* **37**, 4631-4636 (2017)].
- 2) vdW layer GeSe:
[*ACS Nano* **16**, 1308-1317 (2022)].

The evidence extracted from the abovementioned references is listed in **Figure R7** below.

Figure R7. Experimental evidence of irreversible AFE–FE transition in literature.

The electric field-driven AFE–FE transition emerges at the atomic scale and involves many complex collective interactions, including structural distortion, atom displacement, covalent bond breaking/forming, et al. If any of these affected factors are irreversible (for example, domain formation and reorientation), the whole relaxing process may be inhibited. Therefore, the existence of a stable FE state induced from AFE is theoretically reasonable, as demonstrated experimentally by different groups.

We present both theoretical and experimental evidence for verifying the CuCrP_2S_6 FE stability. Meanwhile, we note that although CuCrP_2S_6 -FE-based memories have been intensively studied in recent years, the mechanisms of their phase transition, which includes interactions between the structural transformations, lattice strain, and FE domain formation/reorientation, remain not fully understood. Therefore, along this line there is a large space for future work to explore their physics.

SHG measurement

We probed ferroelectricity using the second harmonic generation (SHG) (**Figure R8a**). Under the excitation of 1064 nm laser pulse, the poled CuCrP_2S_6 flake generates a strong SHG signal (Figure R8b), indicating the broken inversion centrosymmetry associated with ferroelectricity, which provides strong evidence for the presence of ferroelectricity. Figure R8b shows that the SHG signal is stable and persistent without

changing over time. 4 hours later after poling the crystal, the peak intensity of the SHG signal remains strong without any decay.

We observed strong SHG signals in the poled flakes with thicknesses varying from 100 nm to 4 nm (Fig. R9). The polar plot of the SHG intensity measured on a 10-nm flake 4 hours later after poling is shown in Figure R8c. The result shows six-fold symmetry, which agrees well with the hexagonal symmetry in CuCrP_2S_6 crystal structure. We note that the successful detection of SHG signals another important evidence for the stability of ferroelectricity.

Figure R8. SHG measurement. (a) Schematic of SHG measurement. (b) Typical SHG peaks on CuCrP_2S_6 for 5 minutes and 4 hours later after poling. (c) Polar plot of SHG intensity for poled CuCrP_2S_6 flakes.

Figure R9 displays thickness dependent SHG intensity mapping of CuCrP_2S_6 flakes. A thicker sample exhibits a stronger SHG signal. This observation provides clear evidence of a direct correlation between the sample thickness and the SHG intensity. The decreased SHG responses may be attributed to the substrate clamping effect, which is consistent with the previous reports of thickness dependent PFM responses [*Nat. Commun.* **7**, 12357 (2016); *ACS nano* **12**, 4976-4983 (2018)].

Figure R9. Mapping of SHG intensity. Insets: optical images of the flake.

Breakdown concern

Concerning the breakdown of the crystal under a high voltage, we found that the crystal collapses when suffering from a high voltage beyond the coercive field for a long time. This breakdown is caused by the long-range migration of Cu ions from the octahedral sulfur framework where they originally reside, thus showing a delay effect upon poling the crystal. In our experiment, the breakdown voltage is around 10 V/10 nm (**Figure R10a**) and the breakdown will most probably occur after applying this voltage for more than 5 min. After the breakdown, the device exhibits a short-circuit behavior and loses resistive-switching characteristics (**Figure R10b**). To avoid the breakdown, the working voltage was strictly limited to 1 V/10 nm and below for DC tests, and the pulse length was limited to 3 s and below for AC tests. We ensured that the crystal is not subject to the breakdown condition, allowing for safe and reliable switching operation without causing any permanent damage to the device.

Figure R10. (a) Device breakdown test. (b) I-V characteristics of device before and after breakdown.

To accommodate the suggestions, we have added SHG measurement and device breakdown test results in the revised manuscript.

Response to Reviewer #2

The authors are pursuing resistive switching functionality in the emerging class of transition metal thiophosphates. This family of materials has yet to deserve the same level of attention as structurally simpler TMDs, and finding new functionalities is a great way to motivate interest in the topic. However, the authors are attempting to connect the observed resistive switching in CuCrP_2S_6 to supposed antiferroelectric activity. And it is in this respect that the paper falls short. In my opinion, the presented data cannot be considered as evidence for antiferroelectricity. Meanwhile, resistive switching is reasonably robust. If the authors can find a way to explain RS without polar transitions, or support the existence of transitions in a more definitive way, the paper can be reconsidered for the publication.

Specifically:

(1) The simplest argument why polarization activity cannot be the reason for observed RS properties is that the transition temperatures for CuCrP_2S_6 are in the sub 200K range. A very nice recent study (<https://onlinelibrary.wiley.com/doi/full/10.1002/adfm.202204214>) of this material clearly shows this. Also earlier reports on CuCrP_2S_6 report low transition temperature (<https://doi.org/10.1016/j.ssc.2005.05.040>). Meanwhile, present studies were carried out at room temperature and above.

Thank you for raising concerns about our observed resistive switching (RS) phenomenon. We sincerely appreciate the important references suggested in the comments. Actually, we have also noticed these peer works, which bring the significance of ferroelectricity and antiferroelectricity in transition metal thiophosphates into researchers' attention. We were greatly inspired by these papers and, therefore, utilized CuCrP_2S_6 to construct the memristors for neuromorphic computation.

Our wide-range temperature-dependent dielectric characterization suggests remarkable phase transitions at 195 K and 470 K, providing new insights for deciphering the CuCrP_2S_6 polar properties. **These multiple phase transitions suggest that polarization switching can emerge at room temperature or above and govern the observed resistive-switching (RS) phenomena.** For more details, please see the following content.

As shown in **Figure R11**, we acquire the low-temperature (below room temperature; Figure R11a) and high-temperature (above room temperature; Figure R11b) dielectric constant using broadband dielectric spectroscopy (Novocontrol Technologies). The low-temperature dielectric curves show two transition points at 140 K and 195 K, which

coincides with the results reported in the references suggested in the comments (**Figures R12 and R13**). Importantly, we discovered a pronounced peak at 470 K in high temperature range (Figure R11b), above which polar CuCrP_2S_6 eventually transitions to a paraelectric crystal. This unprecedented discovery suggests that CuCrP_2S_6 dipoles do not vanish above 195 K.

We would also like to draw your attention to the fact that ferroelectric materials can exhibit first, second, third, and even more transition points when they evolve from ordered to disordered states, as referenced in the articles ([1] *Phys. Rev.* **93**, 672 (1954); [2] *Appl. Phys. Rev.* **4**, 041305 (2017); [3] *J. Am. Chem. Soc.* **139**, 8752-8757 (2017); [4] *Nat. Commun.* **12**, 284 (2021)). For the CuCrP_2S_6 crystal, the temperatures at $T = 140$ K and $T = 195$ K are undoubtedly first and second phase transition points for ferroelectric coupling. Our measurement in Figure R11a shows the same curves as these early works. The dielectric constant exhibits a sharp rise at $T=140$ K followed by a gradual decrease until $T=195$ K, across all tested frequencies.

However, if CuCrP_2S_6 completely loses polar ordering after undergoing these two low-temperature phase transition points, the dielectric constant should dramatically decrease at $T > 195$ K. Nevertheless, we find that another peak appears above room temperature at $T = 470$ K, indicating the presence of a higher transition point for ferroelectric phase transition. These results indicate that the current understanding of this material in the scientific community is still in its infancy, and it would be premature to assert that the ferroelectric coupling completely disappears above $T = 195$ K. Therefore, we believe that ferroelectric coupling still exists above 195 K, and we conducted further investigations to explore the mechanisms underlying this intriguing phenomenon.

Figure R11. Temperature dependence of CuCrP_2S_6 dielectric permittivity measured at $f = 20$ kHz, 50 kHz, 100 kHz, 200 kHz, and 500 kHz under (a) low temperature and (b) high temperature.

[*Adv. Funct. Mater.* **32**, 2204214 (2022)]

cooperative (antipolar) behavior sets in. Also, no Curie-Weiss law is found throughout the investigated T-range. The nonpolar phase does not appear to be of the usual paraelectric type, most probably because of relatively strong dipole-dipole interactions even at temperatures above the transitions.

[*Ferroelectrics* **185**, 135-138 (1996)]

Figure R12. Dielectric permittivity of CuCrP₂S₆. (a) is adopted from [*Adv. Funct. Mater.* **32**, 2204214 (2022)]. (b) is adopted from [*Ferroelectrics* **185**, 135-138 (1996)].

[*Solid state communications* **136**, 173-176 (2005)]

Figure R13. The heat capacity of CuCrP₂S₆ due at low temperature (2 K to 350 K), adopted from [*Solid state communications* **136**, 173-176 (2005)].

Figure R14. DSC measurement of CuCrP_2S_6 at high temperature (320 K to 550 K).

Nat. Commun. **12**, 284 (2021).

J. Am. Chem. Soc. **139**, 8752-8757 (2017)

Figure R15. DSC curves for ferroelectric phase transitions adopted from references.

We conducted the differential scanning calorimetry (DSC) study using calorimeters (DSC-TA Discovery 250) to validate the conclusions drawn from the dielectric constant measurements. As shown in **Figure R14**, the DSC curve clearly exhibits two endothermic peaks at 386 K and 470 K for the heating run. These peaks are in perfect agreement with the critical temperatures obtained from dielectric measurements. We note that very similar DSC curves (**Figure R15**) have been reported for hybrid perovskites [*Nat. Commun.* **12**, 284 (2021) ; *J. Am. Chem. Soc.* **139**, 8752-8757 (2017)]. Overall, our findings through the calorimetric study provide critical new results to complement what were missed in previous studies, which were limited to low temperatures (Figure R13) [*Solid state communications* **136**, 173-176 (2005)], leaving no doubt that CuCrP_2S_6 ferroelectric phase transitions occur above room temperature and CuCrP_2S_6 exhibits ferroelectric order at room temperature.

Figure R16. Temperature dependent SHG measurement.

We conducted temperature-dependent second harmonic generation (SHG) measurement, and found that the results (Figure R16) are consistent with dielectric constant and calorimetric measurements. As the temperature increases above ~ 500 K, the SHG intensity significantly decreases. It is noted that the transition temperature obtained from SHG measurement is slightly higher than that obtained from the dielectric constant measurement. This may be attributed to the delayed temperature equilibrium caused by the insufficient thermal conductivity of the substrate, because temperature is set at a fixed ramping rate of 10 K/min during the SHG measurement.

Overall, the results from dielectric, DSC, and temperature-dependent SHG measurements finely coincide with each other, confirming the critical phase transition points that have not been previously reported in high temperature range. These interesting results suggest the existence of more complex and intriguing phase transition dynamics in this emerging 2D material. Our further work is to establish a reliable physical model to better describe the driving mechanism, and to create clearer physical pictures for the phase transitions.

(2) Scanning probe microscopy such as PFM is a notoriously poor technique to detect polarization and ferroelectric properties in materials without well-defined domain structure (<https://aip.scitation.org/doi/10.1063/1.4979015>). This is certainly the case in present study. The images in Figure 5 reveal obscure boundaries of poled regions, that are also "streaky" in the direction of the tip scanning. There are ways to extend PFM, using a combination with KPFM. However, to prove polarized structures, the best measurement would be second harmonic generation. Both on pristine and modified areas.

We agree with your point that some artifacts may be introduced by PFM measurement. In order to verify PFM data, we conduct second harmonic generation (SHG) (**Figure R17**) and KPFM measurements as you pointed out. Under the excitation of 1064 nm laser pulse, poled CuCrP_2S_6 flakes generate a strong SHG signal (Figure R17b), indicating the broken inversion centrosymmetry associated with ferroelectricity existing in the poled crystals, which provides compelling evidence for the presence of ferroelectricity. Electrically poled region exhibits a stronger intensity than that of the pristine region. The polar plot of the SHG intensity measured for both pristine and poled sample is shown in Figure R17c. Pronounced six-fold symmetry agrees with the hexagonal symmetry viewed along c-axis direction in CuCrP_2S_6 crystal structure. In addition, **Figure R18** shows the enhanced SHG signal of CuCrP_2S_6 after poling the crystal by the bottom electrode. Figure R18a illustrates the schematic of our experimental setup where a CuCrP_2S_6 flake was transferred onto the bottom electrode connected to a voltage source. As shown in Figures R18b and c, after the application of +6 V on the bottom electrode, the portion above the electrode generates an enhanced SHG intensity. The greatly enhanced SHG intensity for poled (modified) crystal reveals the effect of electric-field induced AFE-to-FE transition. Additionally, Figure R16 depicts the temperature dependence of SHG intensity, also indicating that the SHG signal is ferroelectricity-associated.

Figure R17. SHG measurement. (a) Schematic of SHG measurement. (b) SHG peaks on CuCrP_2S_6 at 532 nm for modified and pristine flakes. (c) Polar plot of SHG intensity for modified and pristine

flakes.

Figure R18. (a) The illustration of SHG measurement setup. The CuCrP_2S_6 flake was transferred onto the bottom electrode connected to the voltage source. The red boxes mark the SHG mapping area. (b,c) SHG intensity mapping at the positions marked in (a).

Figure R19. Surface potential mapping measured using SKPM.

As the reviewer mentioned, there is a direct correlation between polarization states and their surface potentials. Regions with aligned polarization tend to electrostatically induce positive/negative charges over the interfaces, leading to variations in the surface potential. In Figure R19, the local surface potential image for both pristine (outside the square) and poled (inside the square) areas are presented, revealing a clear distinction between them. The surface-potential difference of the modified area is found to be 500 meV higher than that of the unpoled area. This scanning kelvin probe microscopy (SKPM) measurement provides additional information that confirms our PFM results.

To accommodate the suggestions, we have added SHG measurement results to the

revised manuscript and the Supplementary Materials.

(3) The authors never consider ionic conductivity as the origin or RS. However it is most likely what's causing the observed resistive effects. Generally I expect that any field-induced processes reported in this paper will strongly change the chemical structure (e.g. topography in Fig. 5). The authors should either provide robust evidence to the contrary, or embrace structural change as the source of RS and thoroughly revise the hypothesis and explanation of their work.

We appreciate you for proposing a plausible explanation for the RS. We fully agree that that ion conduction/migration might exist in the material. However, as previously reported [*Nat. Mater.* **19**, 43-48 (2020); *Materials Horizons* **7**, 263-274 (2020)], **ferroelectric switching in ferroelectric conductors, such as CuInP₂S₆ and CuCrP₂S₆, is always accompanied by ionic conductivity.** Combined with our experimental observations, **we believe that ferroelectricity and its related ionic conductivity in CuCrP₂S₆ crystals are not mutually exclusive, and together contribute to the RS phenomena.** The structural change as concerned only emerges when large electric fields are applied for a long time (> 10 minutes).

First, we found that surface morphology is altered when a sufficiently large voltage was applied to the PFM tip over a sufficient time (**Figure R21**). This result was obtained by using an exceptionally slow scanning manner, which corresponds to 0.25 Hz per line and takes up 17 minutes for each frame. In contrast, if the scanning time of each frame is limited to be less than 10 minutes, the change in morphology could not be observed. In total, only 10 % (5 among 50 flakes) exhibit this phenomenon with high voltages and slow scanning, which is in line with the assumption for ion migration. However, in our device test, the applied DC voltages are limited to 1 V to avoid obvious structural changes.

(1)

Van der Waals layered CuInP₂S₆ (CIPS) is an ideal candidate for developing two-dimensional microelectronic heterostructures because of its room temperature ferroelectricity, although field-driven polarization reversal of CIPS is intimately coupled with ionic migration, often causing erratic and damaging switching that is highly undesirable for device applications. In this work, we develop an alternative switching mechanism for CIPS using flexoelectric effect, abandoning external electric fields altogether, and the method is motivated by strong correlation between polarization and topography variation of CIPS. Phase-field simulation identifies a critical radius of curvature

(2)

The hallmark of ferroelectricity is spontaneous polarization that can be switched by an external electric field (22, 23), while in CIPS, the polarization switching is intimately coupled with ionic migration (13, 20, 21). This is because paraelectric-ferroelectric phase transition and polarization switching are driven by hopping of Cu ions between two equivalent positions in the S₆ octahedron of CIPS (24), and such hopping is also believed to result in the onset of ionic migration (13). The activation energies for ionic conductivity of CIPS

[*Sci. Adv.* **8**, eabq1232 (2022)].

Figure R20. (a) Schematic crystalline structure of CIPS and migration of Cu ion coupled with polarization switching under a scanning probe. (b) Schematic flexoelectric effect. (c) Domain evolution associated with topography variation as revealed by vertical PFM amplitude and phase mappings, both imposed on 3D AFM topography. This figure was adopted from [*Sci. Adv.* **8**, eabq1232 (2022)].

Figure R21. Left: CuCrP_2S_6 topography change after writing a square pattern by applying +15 V tip bias in the center with a very slow scan rate (0.25 Hz per line, 17 min per frame). Right: no topography change with a fast scan rate.

It is important to note that the observed surface morphology changes are not contradictory to the presence of ferroelectricity, which can be seen in previous works. For example, ref. [*Sci. Adv.* **8**, eabq1232 (2022)] reports that ion migration, ferroelectricity, and ferroelasticity can coexist, couple together, and energetically support each other in CuInP_2S_6 (**Figure R20**). The presence of ion migration/conductivity is prompted by ferroelectric switching, and does not exclude the significant role of ferroelectricity in RS. We also refer to the latest important reference proposing the application of ion migration as a memory medium [*Nat. Commun.* **13**, 574 (2022)]. This article demonstrates the electrical-scan-rate dependence of ion migration, providing us with important insights. Following this paper, we attempt to confirm the significant role of ion migration in our device.

DISCUSSION

Polarization switching is one of the major bottlenecks for the applications of CIPS that use its ferroelectricity, as the electrically driven process often causes ionic migration and damages, and the subtle balance has to be carefully maintained such that the applied electric field reverses the polarization without triggering longer-range ionic redistribution. Mechanical switching based on flexoelectric effect can effectively avoid such problem, enabling polarization reversal without inducing ionic migration. Notice that strain gradient can be

[*Sci. Adv.* **8**, eabq1232 (2022)].

We conducted control experiments on devices under different scanning rates. To create the device with enhanced ion migration, we treated CuCrP_2S_6 flakes using PFM tip with a large voltage for a long time. This process resulted in the extraction of localized ions, enforcing them to break their original bonding inside the unit cell and become freely mobile throughout the material. The devices with suppressed ion migration were fabricated using untreated CuCrP_2S_6 flakes. For the devices with enhanced ion migration, slow scanning leads to larger currents and larger switching windows (**Figure R22a**), but the phenomenon is less obvious under fast scanning (**Figure R22b**). This result is consistent with the observations in [*Nat. Commun.* **13**, 574 (2022)]. As we

expected, the device without ion migration does not exhibit a scan rate dependence. This discrepancy suggests an effective approach to distinguish whether only ion migration plays a role in the device. The devices reported in our manuscript do not exhibit any scanning speed dependence, indicating that ion migration is suppressed.

Meanwhile, it is worth noting that the devices with ion migration collapse shortly during the test. The surface morphology of the treated sample changed significantly in the device with ion migration, making it more susceptible to device breakdown and rendering it unsuitable as a functional material for RS devices.

Figure R22. Scan rate dependent I–V characteristics of device made from (a) enhanced ion-migration and (b) pristine CuCrP_2S_6 flakes. Enhanced ion-migrated flakes were scanned using PFM probes before fabricating the devices.

Temperature-dependent electrical hysteresis rules out ion-related filamentary switching. We considered the influence of temperatures in CuCrP_2S_6 performance. In general, most of metal ions in transition metal thiophosphates require additional energy to break the original bonds and get rid of the unit cell for free migration, so in principle the ion migration-based RS device should exhibit an enhanced performance at high temperatures. For example, **Figure R23** displays this trend in Na^+ intercalated and oxide MoS_2 . By contrast, combining our above observations on the temperature dependence of SHG intensity (Figure 16), dielectric spectra (Figure R11), DSC curves (Figure R14), and I–V characteristics (Figure 6 in the main text), it is noticed that both

RS performance and ferroelectricity simultaneously decay with increasing temperatures. These findings demonstrate that CuCrP_2S_6 RS is primarily dominated by ferroelectricity, rather than ion-related conductive filaments.

Nano Lett. **21**, 10400-10408 (2021). *Nat. Electron.* **1**, 130-136 (2018)

Figure R23. Temperature dependent electric characteristics of ion migration-based memristor, adopted from [*Nano Lett.* **21**, 10400-10408 (2021); *Nat. Electron.* **1**, 130-136 (2018)].

High-temperature treated devices also rule out the impact of ion-related filament switching. It is widely known that ion migration contributes to RS by forming ionic conductive filaments (Figure R24c). RS tends to exhibit abrupt changes upon filament formation. We found that a minority of devices exhibit such behavior after annealing. As shown in **Figure R24a**, at a voltage of 0.6 V, the current abruptly increases, indicating the formation of a filament. The ions responsible for filament formation are likely copper ions that are decoupled from their binding sites due to thermal activation. Interestingly, after cooling the device back to room temperature, applying a negative voltage of -1.2 V causes a sharp drop in current (Figure R24b), suggesting the erasing of the ionic conductive filaments. In contrast to the abrupt changes induced by conductive filaments, the RS reported in our manuscript exhibits slow, gradual changes in the I-V curves, forming a clear contrast to the abrupt changes caused by conductive filaments.

Figure R24. (a) Filament forming at high temperature. (b) Filament erasing after cooling back. (c) Schematic of filament forming and erasing.

Based on the aforementioned elucidation, we would like to highlight that ferroelectricity and its related ion conductivity are not two mutually exclusive. Instead, they work together to govern the RS in CuCrP_2S_6 memristor. We appreciate the suggestion of ion conductivity and are pleased to include the discussion on this point in the revised manuscript. Under some circumstances (e.g., high-temperature annealing), we believe that copper ion conductivity can transform into electrical filaments and induce filamentary RS. However, considering temperature-dependent I-V hysteresis, PFM, and SHG results in CuCrP_2S_6 devices, we strongly believe that ferroelectricity dominantly contributes to RS. We added the above data and discussion over ion conductivity in the revised manuscript.

Response to Reviewer #3

Ma and coworkers report on the electric field induced ferroelectric phase transition in $\text{CuCr}_2\text{P}_2\text{S}_6$ (CCPS) and the integration of such van der Waals materials in a capacitor design for memristive type application. The higher ferroelectric transition temperature, compared to existing van der Waals systems allows for broader temperature range for applications. The authors demonstrate a high ON/OFF ratio in the resistance change controlled by the ferroelectric polarization reversal.

Controlling the ferroelectric domain population allows for beyond binary response and hence for neuromorphic type applications.

I find the manuscript interesting, the push for energy efficient electronic devices is in my opinion of high interest. Here, in particular I appreciate the direct demonstration of the controlled resistance states driven by the domain population, directly imaged by scanning probe microscopy.

I would recommend this work for further consideration once the authors address the following points:

Thank you for your positive comments on the significance of our work. Your comments are very helpful to improve the quality of our revised manuscript. We have carefully considered all comments and have made the necessary revisions to address the concerns. We provide a detailed point-by-point response below, and hope that our revisions meet your expectations.

- Several sections of the manuscript may appear confusing to non experts. In the introduction, the “Cu atoms in disorder” may need further disorder and context.

Thank you for this helpful suggestion. Regarding the confusing points, we have revised the sentences in the manuscript as follows.

However, because of the low Curie point ($T_c = 42 \text{ }^\circ\text{C}$) of CIPS, high temperatures of $>42 \text{ }^\circ\text{C}$ destroy the ferroelectric order and induce a transition to the paraelectric order. Consequently, the performance of these memory devices certainly deteriorates.

- The poor thermal stability may not be correlated to the ferroelectric transition temperature. Could the authors be more specific? Do they address the chemical stability (e. g. degradation of the structure) at elevated temperature or, specifically the increase of ferroelectric T_c ?

We appreciate this comment that motivates us to conduct an in-depth investigation into ferroelectric transition temperature and consider chemical stability.

We performed a thorough investigation for characterizing the ferroelectric transition temperatures. As shown in **Figure R25**, the first and second phase transitions occurring around 380K and 470K are indicated by the differential scanning calorimetry and dielectric permittivity curves. Therefore, we consider the impact of the atmospheric environment on the chemical stability of the material and propose an approach of encapsulating the memristor with hexagonal boron nitride (h-BN). h-BN is chemically inert and stable, making it an ideal material for protecting 2D devices from the surrounding environment. The fabrication processes are schematically illustrated in **Figure R26a** and the as-encapsulated memristor is presented in Figure R26b. The electric characteristics are found to be more stable after encapsulation, as evidenced by the I-V curves that remained unchanged after exposure to air for a month. However, there is no obvious improvement in the temperature stability. This result aligns with the fact that the device's chemical stability can be elevated by encapsulation, but limited by phase transition temperature.

Figure R25. (a) Temperature-dependent DSC curve for a CuCuP_2S_6 crystal and (b) temperature-dependent dielectric permittivity curves measured under different frequencies for a CuCuP_2S_6 crystal.

Figure R26. (a) Fabrication process of the encapsulated memristor. (b) Optical image of the as-fabricated encapsulated memristor.

- The term of “giant” polarization is no suitable here. Report on giant polarization refer to values exceeding 150 micro/cm². (Phys. Rev. Lett. 107, 147602 (2011)).

Thank you for pointing out this problem. We substituted “giant” as “larger polarization compared with CIPS”.

The polarization of 2D layered ferroelectricity, due to the vdW-gap-weakened interlayer dipole coupling, is not comparable with ferroelectric oxides. Ferroelectric oxides grown on specific templates can be further enhanced by the strain.

- Was the polarization of CCPS measured? Using PUND? Could the authors show a

P-E loop?

We characterize poled crystals using a ferroelectric tester (aixACCT TF Analyzer 2000, Germany). We fabricated a ferroelectric capacitor by patterning electrodes using electron beam lithography (EBL) on a 50-nm CuCrP_2S_6 flake. After poling the flake with a DC voltage of 2 V, we obtained its polarization–voltage (P–V) curves. The result shows pronounced hysteresis behavior (**Figure R27**).

Figure R27. Polarization–voltage hysteresis (P–V) loop measured by ferroelectricity tester on CuCrP_2S_6 ferroelectric capacitor. Test frequency: 0.1 Hz.

Following the suggestion, we have added P–V loops in the revised manuscript.

- The authors attribute the change of resistance to the reversal of polarization and use the Schottky emission and FN tunneling as main mechanism for the transport. Can't the authors consider tunneling electroresistance? Here the asymmetric tunneling potential and tunneling electroresistance may explain as well the polarization dependent resistance change?

We adopted FN tunneling instead of tunneling electroresistance (TER) as an explanation because the thickness is not suitable for ordinary tunneling.

TER is applicable when the thickness gets thinner (e.g. below 10 nm), because TER relies on ordinary tunneling, which has a distance limit due to the finite penetration depth of electron wavefunction. In general, the maximum allowed tunneling distance is $d=4$ nm. According to the quantum tunneling model, the possibility of tunneling is proportional to $e^{-2\kappa d}$ (the derivation is shown below), which means that the possibility of tunneling decreases exponentially as the barrier thickness increases (**Figure R28**). Considering that our sample has a thickness of approximately 10 nm, exceeding the

typical tunneling thickness, therefore, we employ the FN tunneling model as an explanation.

$$\Psi_{incident}(x) = Ae^{-kx} + Be^{kx}$$

$$\Psi_{barrier}(x) = Ce^{-kx} + De^{kx}$$

$$\Psi_{tunneling}(x) = Fe^{kx}$$

$$|\Psi_{tunneling}(x)|^2 = |F|^2 = |C|^2 e^{-2\kappa d}$$

$$\text{where } \kappa = \sqrt{\frac{2m}{\hbar^2} (V_0 - E)}$$

Figure R28. Ordinary quantum tunneling model for free electrons.

We explore the possibility of TER occurring in ultrathin flakes by fabricating a memristor with smaller thicknesses (<4 nm), which meets the requirements for ordinary tunneling to occur. Such ultra-thin materials are highly susceptible to damage from metal deposition, which can frequently cause short circuits in the device. To address this issue, we replace the top electrode with graphene to form a non-destructive contact. **Figure R29** shows this ultra-thin memristor we fabricated, which exhibits almost no hysteresis in its I-V curve. This observation suggests that, although tunneling possibility is facilitated in ultra-thin ferroelectrics, their ferroelectricity may be lost due to the enhanced depolarization field.

Figure R29. Ferroelectric junction based on ultrathin (4 nm) CuCrP₂S₆ with graphene top electrode and metal bottom electrode.

I would recommend the authors to refer to the studies Nature 460, 81 (2009), Science 313, 181–183 (2006) among many others.

Thank you for recommending the valuable references. Ref. [*Nature* 460, 81 (2009)] reports the approach of probing ultrathin polarization (down to 1 nm) by measuring the TER using CAFM. We have added some discussion over this reference. Ref. [*Science* 313, 181–183 (2006)] is a perspective article, which discusses the prospects of combining ferroelectric tunneling resistance and spin tunneling resistance to expand the application of ferroelectric materials in novel electronic devices. In fact, we are also very interested in this research topic. It is worth mentioning that CuCrP₂S₆ also exhibits antiferromagnetic property at extremely low temperatures ($T < 30$ K). We are looking forward to studying the tunneling effect induced by multiferroicity in our material in our next projects.

- Here, controlled domain population also leads to several levels of resistance and studies Nature Commun. 13, 3159 (2022) may deserve some reference too in the context of the work presented.

Thank you for providing this important reference. This reference provides valuable insights into the physics mechanism behind multilevel polarization switching in ferroelectric materials. The author revealed that multilevel polarization switching is based on the competition between different crystal structures in ferroelectric materials near the morphotropic phase boundary (MPB). Under strain and DC voltages, there are many small domains with the polarization direction oriented in opposite directions. They do not affect each other's polarization direction. This allows for multilevel polarization switching. The phase competition near the MPB, in combination with epitaxial strain, increases the responsiveness to external stimuli.

We cited this reference in the revised manuscript as an important theoretical support.

- Ideally a thickness dependence of the ferroelectric material in transport properties may help identifying the mechanism. Have the authors tried different thicknesses? How do the I/V curve evolve as the films gets thicker?

Thank you for the constructive suggestion. We have collected a set of I–V curves (**Figure R30**) for samples with varied thicknesses. The resistive switching on/off ratio is enhanced as the sample thickness decreases. Intuitively, a decrease in thickness down to nanoscale brings the sample closer to the critical thickness of ferroelectricity and thus a larger depolarization field that can deteriorate the ferroelectric behavior. Nevertheless, the reduction in thickness still leads to an improvement of device performance, which implies that the source of resistive switching is not only based on interface effects. This is the reason that we proposed the FN tunneling model as a supplement to the well-established interfacial ferroelectric-modulated Schottky emission model.

Figure R30. I–V characteristics for samples with varying thickness.

Following the suggestions, we included the above results in the revised manuscript.

- The PFM data in 1f seem to show 3 contrast levels, which do not appear later in figure 5.

The PFM in Figure 1f in the manuscript is due to the PFM tip shift. The tip shift is relatively small in Figure 5, and therefore, appears less visible.

Because 2D material flakes are relatively fragile and soft, we utilized PFM conductive probes with lower stiffness (spring constant: 2 N/m), which could cause a probe shift when encountering a sharp topographic variation during the scanning process, such as dust and particles adsorbed on the surface of sample due to electrostatic adsorption, leading to some unavoidable contrast deviation. Here, we present additional PFM data (**Figure R31**) with uniform contrast to demonstrate the reproducibility of our experimental results.

Figure R31. Additional piezoresponse phase images of other two samples.

Could the authors show the time dependence of the phase or alternatively try to optimize the AC reading bias to get sharper contrast?

A larger AC reading bias can certainly give rise to a sharper contrast. However, we set the read voltages at a relatively smaller value (less than 0.8V) to avoid the artifacts. In contrast to the oxide films with strong hardness, the surface of CuCrP_2S_6 is relatively soft and fragile, making it susceptible to morphological changes when subject to large AC scanning voltages. A previous study has shown that large AC voltages applied on the PFM tips can cause domain wall movement, distortion, and reorientation, as well as ion migration-induced damage (**Figure R32**, adopted from [*Sci. Adv.* **8**, eabq1232 (2022)]).

To achieve a trade-off between sharper contrast and the reduction of destructive damage, we limited the amplitude of the applied voltage. Furthermore, we employed additional characterization techniques, such as second harmonic generation (SHG) testing (**Figure R33**), dielectric constant measurements (Figure R25b), and calorimetry (Figure R25a), ferroelectric tester (Figure R27), to supplement and support PFM results for confirming the ferroelectricity. Additionally, time dependence of the PFM phase is shown in **Figure R34** and added in the revised manuscript.

very similar to those previously reported in literature (3). During PFM measurement (37–42), a small AC voltage is applied through a scanning probe, exciting piezoelectric vibration of CIPS that can be measured by the photodiode. In principle, it is also possible to apply a larger DC voltage to switch the spontaneous polarization, as schematically shown in Fig. 1A, which inevitably induced migration of Cu ions under the scanning probe, and often causes sample damages (3, 13, 20, 43). When a DC voltage of ± 6 V is applied through the scanning probe, substantial topographic change is observed in Fig. 1 (D and E) due to ionic migration–induced damage, while it fails to switch the polarization as shown by fig. S1. In other

Figure R32. Large tip voltage-induced ion migration, adoped from [*Sci. Adv.* **8**, eabq1232 (2022)].

Figure R33. (a) The SHG peak on a poled CuCrP_2S_6 flake. The excitation laser is a 1064 nm infrared pulse laser at 5 mW, and the strong SHG emission at 532 nm is observed. (b) Polar plot of SHG intensity.

Figure R34. PFM phase images measured with 5-minute (a) and 24-hour (b) delays after writing the box-in-box pattern.

REVIEWER COMMENTS

Reviewer #1 (Remarks to the Author):

I have read thru the authors responses to my comments. While it is a step in the right direction, I have to say I am not fully convinced, especially regarding the P-E loops they present in the response. Those "loops" do not look like a normal ferroelectric loop. They should check it carefully.

Regarding the PFM images: all antiferroelectrics will show a ferroelectric like behavior when a field is applied to them. This will relax back, as the PFM images suggest.

The authors should properly caveat the text to make sure that the data is not misleading

Reviewer #2 (Remarks to the Author):

The authors admittedly carried out a lot of additional measurements to support their hypothesis. While I still hold the opinion that ferroelectric transitions are not directly involved in memristive switching, there is enough evidence in the paper to at least consider this scenario. The revised manuscript is significantly improved. I would like the authors to provide response to these minor comments:

(1) The P-E loops in Fig. S7 look like "classical banana ferroelectrics" - i.e. leaky dielectrics rather than ferroelectrics. It would be helpful to provide: (a) P-E loops BEFORE poling (presumably pointing to non-polar or double-hysteretic antiferroelectric state); (b) I-V curves of these devices in the measured voltage range.

(2) SHG is a welcome addition to the paper. However, From both S4 and S5, SHG intensity is observed EVERYWHERE across the flake, not only the poled regions. (a) The authors should clearly show the outlines of the poled region in S4. (b) The scalebar in S5 is 10^7 , while it is only 10^3 or so in S4. What is the origin of the discrepancy? (c) What happens in S5 after reversing pulse polarity to -6V?

(3) None of the I-V curves shown have threshold behavior characteristic of ferroelectric switching. There are instead smooth hysteresis loops, more aligned with charge-trapping/detrapping scenario. What specific measurements rule out the electret-type response of these films, which does not require phase transitions of any sort?

(4) Fowler-Nordheim tunneling model is temperature-independent, while the experiments clearly are. This should be reconciled. Also, all parameters of FN fitting should be specified.

Reviewer #3 (Remarks to the Author):

The authors have addressed all the points raised during the previous step.

I would gladly recommend it for publication once the last point dealing with the optical SHG characterization is considered:

The SHG intensity as function of wavelength, as shown in Figure R33 has limited value, since any surface breaks inversion symmetry and hence emits SHG, the plot is not a proof for a polar behavior.

Can the anisotropy plot be fitted considering the ferroelectric point group symmetry of the material? Alternatively, a drop of SHG at the ferroelectric T_c would suffice (PHYSICAL REVIEW LETTERS 123, 147601 (2019)).

Response to Reviewer #1

I have read through the authors responses to my comments. While it is a step in the right direction, I have to say I am not fully convinced, especially regarding the P–E loops they present in the response. Those "loops" do not look like a normal ferroelectric loop. They should check it carefully. Regarding the PFM images: all antiferroelectrics will show a ferroelectric like behavior when a field is applied to them. This will relax back, as the PFM images suggest. The authors should properly caveat the text to make sure that the data is not misleading

Reply:

Thank you for your positive and critical comments. We optimized the performance of CuCrP_2S_6 ferroelectric capacitors, where the leakage current is largely mitigated. Meanwhile, we employed an up-to-date experimental setup (Radiant Technologies) for measuring polarization magnitude. With these improvements, **a characteristic P–E loop (Figure R1) was obtained.**

Figure R1. P–E loop measured at 100 Hz.

We understand that with time elapsing, the domains will switch back to the ground state (i.e., antiferroelectricity). As the reviewer pointed out, this relaxation process can be confirmed by black dashed line marked in Figure R2.

Figure R2. PFM phase images with written patterns acquired with (a) 5-minute and (b) 1-day delay. Relaxed parts are marked by black dash lines in (b).

Following your suggestions, **we also assessed the relaxation process through surface electric potential mapping**, because the surface potential change is directly associated with the resistance switching behaviors of the memristor. As presented in **Figure R3**, the poled region exhibits considerably high surface potential, while after one week it significantly relaxed to a ground state. This further substantiates the FE–AFE relaxation process.

Figure R3. Retention property for surface potential. Poled regions are marked by dash lines. (a) and (b) were acquired with 5-minute and 1-week delay after poled with the PFM tip, respectively.

We have added these data and related discussion in Pages 3, 23, and 28 of the revised manuscript.

existence of ferroelectricity. The possible reason that explains the observed ferroelectricity rising from antiferroelectric is that the electric field can induce a structural transition from an antiferroelectric to ferroelectric phase^{25,27-29}. With time elapsing, the domains will relax back to the ground state, i.e., antiferroelectricity (Figures S2 and S6).[↵]

regions, confirming the ferroelectricity. Polarization–electric field (P–E) curves were also measured for a 50-nm thick CuCrP_2S_6 flake. Pronounced hysteresis loops (Figure S7) are observed, which unambiguously demonstrates the existence of ferroelectricity.

(Revisions in Page 3)

additional information that confirms the PFM results. The poled region exhibits considerably high surface potential, while after one week it significantly relaxed to a ground state. This further substantiates the FE–AFE relaxation process.[↵]

(Revisions in Page 28)

Response to Reviewer #2

The authors admittedly carried out a lot of additional measurements to support their hypothesis. While I still hold the opinion that ferroelectric transitions are not directly involved in memristive switching, there is enough evidence in the paper to at least consider this scenario. The revised manuscript is significantly improved. I would like the authors to provide response to these minor comments:

Reply:

Thank you for your positive and constructive comments. We have conducted more experiments to address your comments.

(1) The P–E loops in Fig. S7 look like "classical banana ferroelectrics" - i.e. leaky dielectrics rather than ferroelectrics. It would be helpful to provide: (a) P–E loops BEFORE poling (presumably pointing to non-polar or double-hysteretic antiferroelectric state); (b) I–V curves of these devices in the measured voltage range.

Thank you for your important suggestions. For acquiring good P–E loops, we optimized the performance of CuCrP_2S_6 ferroelectric capacitors used for testing P–E loops, and adjusted measurement parameters. Meanwhile, we employed an up-to-date experimental setup (Radiant Technologies) for measuring polarization magnitude. **The results (Figure R4) show a more pronounced ferroelectric coercive field and remanent polarization.** We find that before poling, CuCrP_2S_6 crystals show a negligible hysteresis loop.

Figure R4. P–E hysteresis loops for unpoled and poled samples and the corresponding leakage current measured on the CuCrP_2S_6 ferroelectric capacitor after poling. Poling electric field: (600 kV/cm). Test frequency: 100 Hz.

We have added these data and related discussions in Pages 3 and 29 of the revised manuscript.

domain reversal under the PFM tip-induced electric field. Polarization–electric field (P–E) curves were acquired for a 50-nm thick CuCrP₂S₆ flake. Pronounced hysteresis loops (Figures 1f and S7) are observed, which unambiguously demonstrates the existence of ferroelectricity. The possible reason that explains the observed ferroelectricity rising from antiferroelectric is that the electric field can induce a structural transition from an antiferroelectric to ferroelectric phase^{25,27-29}. With time

(Revisions in Page 3)

(2) SHG is a welcome addition to the paper. However, from both S4 and S5, SHG intensity is observed EVERYWHERE across the flake, not only the poled regions. (a) The authors should clearly show the outlines of the poled region in S4. (b) The scalebar in S5 is 10^7 , while it is only 10^3 or so in S4. What is the origin of the discrepancy? (c) What happens in S5 after reversing pulse polarity to $-6V$?

Reply:

Thank you for raising these questions. We have made clarifications and revisions as below.

- (a) In Figure S4 of 1st-round revision, all surface areas on the CuCrP₂S₆ flake with different thicknesses, were poled by PFM tip. To make it clearer as you pointed out, we marked the outlines of the poled flake and added annotations in the figure captions to clarify this point (**Figure R5**).
- (b) The noted discrepancy primarily arises from the alterations related to the light pathways. Between these two experiments there was a long-time gap, during which we modified the light pathway of optical platform to install accessories and improved the sensitivity/resolution. Besides, other factors such as surface thoroughness, focus distance and integration time are also involved and collectively result in the intensity variation on different samples. The discrepancy will not affect our conclusion, which was based on the comparison of **relative** intensity on different parts of CuCrP₂S₆ crystal under the same experimental condition. To clarify this, we have noted the experimental conditions for these two measurements in the revised manuscript and normalized the intensity for all measurements based on the calibration on standard reference samples, in line with common analysis method [*Adv. Opt. Mater.* **6**, 1701334 (2018)].

Figure R5. Mapping of the normalized SHG intensity on poled flakes. Insets: optical images of the flakes. The outlines of poled flakes are marked by dash lines.

(c) The results from -6 V poling show similar behaviors with $+6$ V poling, because both $+6$ V and -6 V poling facilitate large crystal distortion and thus enhance SHG intensity.

We have revised these data (Figure S4 and S5) in Pages 25 and 26 of the revised manuscript.

Figure S4. Mapping of the normalized SHG intensity on poled flakes. Insets: optical images of the flakes. The outlines of poled flakes are marked by dash lines.↵

(Revisions in Page 25)

(3) None of the I–V curves shown have threshold behavior characteristic of ferroelectric switching. There are instead smooth hysteresis loops, more aligned with charge-trapping/detrapping scenario. What specific measurements rule out the electret-type response of these films, which does not require phase transitions of any sort?

Reply:

Thank you for raising this important question. Actually, threshold-like ferroelectric resistance switching is acquired at specific experimental conditions, i.e., pulse train with alternate poling and read pulses, which is distinct from that of I–V hysteresis loops (i.e., continuous DC voltage sweeping). To address your concerns, we demonstrated threshold-like resistance switching in CuCrP_2S_6 memristors (**Figure R6**), and the behaviors are consistent with previous works (**Figure**

R6.1), which strongly suggests polarization-switching origin.

Figure R6. (a) Resistance hysteresis loop of CuCrP_2S_6 ferroelectric memristor displaying clear voltage thresholds. (b) Voltage pulse train used for obtaining resistance hysteresis loop.

Figure R6.1. Resistance hysteresis loops for ferroelectric tunnel junctions based on (a) BTO/LSMO, (b) BiFeO_3 and (c) CuInP_2S_6 tested by voltage pulse sequences.

Second, ferroelectret is insensitive to temperature variation, whereas both our SHG intensity and RS ratio show degradation upon increasing temperature. This demonstrates that CuCrP_2S_6 flake has ferroelectric nature, rather than ferroelectret.

Figure R7. (a) Temperature dependence of SHG intensity. (b) Temperature dependence of switching ratio of the CuCrP_2S_6 memristor.

As shown in previous revision, we have shown that CuCrP_2S_6 resistance switching mainly stems from polarization switching, rather than charge trapping.

We have added these data (Figure R6 and R7) in Page 4, 14, and 43 of the revised manuscript.

The most striking discovery from our investigation is the observation of a phase transition occurring above room temperature, which is not previously reported. In Figures 1g and 1h, a remarkable drop in SHG intensity is observed around 470 K, suggesting a transition from the ferroelectric to the paraelectric phase. In good

(Revisions in Page 3)

(4) Fowler-Nordheim tunneling model is temperature-independent, while the experiments clearly are. This should be reconciled. Also, all parameters of FN fitting should be specified.

Reply:

Thank you for raising this important question and providing your suggestions. We specified fitting parameters as **Table R1**.

Expression of F–N tunneling:

$$\ln\left(\frac{J}{E^2}\right) = \left(\frac{1}{E}\right) \left(\frac{-4\sqrt{2m^*}(q\Phi)^{\frac{3}{2}}}{3q\hbar}\right) + \ln\left(\frac{q^2}{16\pi^2\hbar\Phi}\right),$$

Fitting curve:

$$y = kx + b,$$

where

$$k = \frac{-4\sqrt{2m^*}d(q\Phi)^{\frac{3}{2}}}{3q\hbar}, b = \ln\left(\frac{q^2l^2}{16\pi^2\hbar\Phi d^2}\right),$$

$$y = \ln\left(\frac{I}{V^2}\right), \quad x = \frac{1}{V}$$

Table R1: fitting parameters for F–N tunneling at investigated temperatures.

Temperature (°C)	k (HRS) (V)	k (LRS) (V)	b (HRS) (A/V ²)	b (LRS) (A/V ²)
27	-4.654	-2.677	-10.581	-12.558
47	-4.487	-2.522	-10.166	-12.131
67	-4.559	-2.494	-9.508	-11.573
87	-4.232	-2.204	-9.325	-11.353
97	-3.900	-1.006	-8.883	-11.777
107	-3.023	-0.338	-9.167	-11.852

Interestingly, while the classic F–N tunneling model is temperature-independent, our observed fitting parameters display a clear temperature dependency. The trend of our results with respect to temperature (**Figure R8**) aligns closely with findings from previous studies (**Figure R9**). As temperature decreases, the temperature dependence almost disappears, likely due to the attenuation of thermal effects.

Figure R8. Temperature dependence of fitting parameters for F–N tunneling.

Figure 5. $\ln(I/V^2)$ versus $1/V$ plot of the data for weakly temperature-dependent $I-V$ data at 180–77 K (regimes II and III in Figure 3). The curves show a transition from direct tunneling to F–N tunneling with a voltage inflection point. (b) Magnified view of the $\ln(I/V^2)$ versus $1/V$ plot, showing the linear behavior between them, in agreement with F–N tunneling.

[Kies, R., et al. "Temperature dependence of Fowler-Nordheim emission tunneling current in MOS structures." ESSDERC'94: 24th European Solid State Device Research Conference. IEEE, 1994]

Fig. 2 : Experimental and theoretical variations of the pre-exponential coefficient A and exponential coefficient B with temperature (F-N current simulation parameters : $\Phi=3.1\text{eV}$, $m_s=m_e$, $m_{ox}=0.5m_e$, $N=10^{20}/\text{cm}^3$, m_e =electron mass).

Figure R9. Temperature dependence of fitting parameters for F–N tunneling in the previous literature.

Here, based on the previous literature, we summarize the reasons for the temperature dependence as follows. (1) Barrier height variation arising from the temperature-induced statistical carrier distribution; (2) Alterations in the band gap induced by thermal fluctuations; (3) Phonon-assisted tunneling. (4) The contribution of current from thermionic emission.

We have added these data (Figure R8 and Table R1) and related discussion in Pages 37 and 38 of the revised manuscript.

The $\ln(I/V^2)$ versus $1/V$ plot is linearly fitted by $y = kx + b$, where $x = 1/V$, $y = \ln(I/V^2)$ and the slopes (k) are extracted for the HRS and LRS at investigated temperatures as shown in Table S1. k is associated with the potential barrier height, which can be

(Revisions in Page 37)

The possible reasons for the temperature dependence are as follows. (1) Barrier height variation arising from the temperature-induced statistical carrier distribution; (2) Alterations in the band gap induced by thermal fluctuations; (3) Phonon-assisted tunneling. (4) The contribution of current from thermionic emission. ←

(Revisions in Page 38)

Additionally, we want to thank you for your previous concerns about transition temperature. You mentioned a nice recent study [*Adv. Funct. Mater.* **32**, 2204214 (2022)], which reports CuCrP₂S₆ ferroelectric phase transition in sub-200 K range. We have cited this work and provided in-depth discussion, with comparison to our newly added evidence (SHG, dielectric curves) identifying the transition at T = 470 K. We believe that these additions are crucial complement and extension for the prior studies on this material.

The most striking discovery from our investigation is the observation of a phase transition occurring above room temperature, which is not previously reported. In Figures 1g and 1h, a remarkable drop in SHG intensity is observed around 470 K, suggesting a transition from the ferroelectric to the paraelectric phase. In good alignment with this observation, our dielectric measurement shows a clear cusp appearing at 470 K in the dielectric curves (Figure 1i), further confirming the occurring of a phase transition around this critical temperature. Remarkably, previous studies identified two phase transitions at $T_{c1} = 145$ K and $T_{c2} = 195$ K^{23,31,32}. Here, we report $T_c = 470$ K as a newly discovered critical temperature for phase transition, above which the dipole coupling completely vanishes. Note that multiple transition points are quite common in ferroelectrics when they evolve from ordered to disordered states³³. Therefore, we believe that T_{c1} and T_{c2} are the first two critical temperatures for the dipole coupling strength alternations, whereas $T_c = 470$ K may be the actual Curie temperature.←

(Revisions in Page 4)

23. Cho K, Lee S, Kalaivanan R, Sankar R, Choi KY, Park S. Tunable ferroelectricity in Van der Waals layered antiferroelectric CuCrP₂S₆. *Adv. Funct. Mater.* **32**, 2204214 (2022).←

35. Moriya K, Kariya N, Inaba A, Matsuo T, Pritz I, Vysochanskii YM. Low-temperature calorimetric study of phase transitions in CuCrP₂S₆. *Solid State Commun.* **136**, 173-176 (2005).←

(Revisions in Page 13)

Response to Reviewer #3

The authors have addressed all the points raised during the previous step.

I would gladly recommend it for publication once the last point dealing with the optical SHG characterization is considered:

The SHG Intensity as function of wavelength, as shown in Figure R33 has limited value, since any surface breaks inversion symmetry and hence emits SHG, the plot is not a proof for a polar behavior.

Can the anisotropy plot be fitted considering the ferroelectric point group symmetry of the material? Alternatively, a drop of SHG at the ferroelectric T_c would suffice (PHYSICAL REVIEW LETTERS 123, 147601 (2019)).

Reply:

Thank you for your positive comments and questions. **As expected, Figure R8 presents a clear SHG-intensity drop appearing at the ferroelectric T_c (around 470 K), which provides evidence for addressing your concern.**

Although SHG signal can be induced by surface broken inversion symmetry, it will not show layer dependence. Yet, our SHG intensity mapping **Figure R9** clearly exhibits layer dependence, verifying our claim that the SHG signal is generated by the broken inversion symmetry of the crystal, rather than solely a surface effect.

Figure R8. SHG intensity reducing at ferroelectric T_c (around 470 K).

Figure R9. SHG intensity mapping on flakes with varying thicknesses.

Regarding the anisotropy plot, the ferroelectric CuCrP_2S_6 belongs to **the space group of P3 (No. 143)**. Thus, SHG intensity can be fitted by $I = I_0 \cos^2(3\theta)$ (Figure R10), where θ is the angle between the incident laser polarization and the a-axis [*Adv. Mater.* **29**, 1701486 (2017)].

Figure R10. Polar dependent SHG intensity fitted by $I = I_0 \cos^2(3\theta)$ according to the space group P3 (No. 143).

We have added these data and related discussions in Pages 4 and 15 of the revised manuscript.

CuCrP₂S₆ crystal. The ferroelectric CuCrP₂S₆ belongs to the space group of P3 (No. 143), consistent with the fact that the SHG intensity can be fitted by $I = I_0 \cos^2(3\theta)$, where θ is the angle between the incident laser polarization and the a-axis³⁰. It is noted that some flakes show strong six-fold SHG intensity even without the history of being applied an electric field, suggesting a potential co-existence of antiferroelectric and ferroelectric coupling. ←

The most striking discovery from our investigation is the observation of a phase transition occurring above room temperature, which is not previously reported. In Figures 1g and 1h, a remarkable drop in SHG intensity is observed around 470 K, suggesting a transition from the ferroelectric to the paraelectric phase. In good

(Revisions in Page 4)

REVIEWERS' COMMENTS

Reviewer #1 (Remarks to the Author):

I read thru this version and it is significantly improved. I would like the authors to add the P-E loop and the PFM data into the main part of the paper.. not as supplementary data. These two pieces of data are central to the premise of the paper and so should not be a supplementary.

After this, the paper is acceptable.

Reviewer #2 (Remarks to the Author):

The authors have once again improved the presentation and interpretation of their results. They clarified most of the questions I raised earlier. I recommend the paper for publication after two following comments are addressed.

(1) One technical detail that I would still like to see is a closer comparison of the ferroelectric and polarization loops. Specifically, I recommend that the authors add an electric field axis in any of the I-V plots in Figure 2. That will bring the comparison closer to Figure 1f.

(2) Are ferroelectric hysteresis loops observed after one or several cycles of resistive switching?

Reviewer #3 (Remarks to the Author):

Based on the improvement of the SHG optical investigation description, the strong temperature dependence observed, the newly added P-E loop, I can now support the present work for publication.

Response to Reviewer #1

I read thru this version and it is significantly improved. I would like the authors to add the P-E loop and the PFM data into the main part of the paper, not as supplementary data. These two pieces of data are central to the premise of the paper and so should not be a supplementary. After this, the paper is acceptable.

Reply:

Thank you for your recognition of the improvement we have made. We deeply appreciate your significant contribution of providing constructive feedback to our paper. We have added the P-E loop and the PFM data into the main part of the paper as you suggested.

(Revised Fig. 1c in Page 15)

the antiferroelectric to ferroelectric phase^{25,27,30-32}. With time elapsing, the domains naturally relax back to the ground state, i.e., antiferroelectricity (Fig. 1c).⁴

(Discussion for Fig. 1c in Page 3)

grown crystal. Grid size: 1 mm × 1 mm. Bottom panel: Raman spectra of CuCrP₂S₆. **c**, Top panels: piezoresponse phase image with a written box-in-box pattern before and after the 1-day relaxation. Relaxed parts are marked by black dash lines. Bottom panel: PFM amplitude (red) and phase (blue)-hysteresis loops for a 10 nm-thick CuCrP₂S₆ flake. **d**, Cross-sectional crystal structures viewed from

(Caption for Fig. 1c in Page 15)

Response to Reviewer #2

The authors have once again improved the presentation and interpretation of their results. They clarified most of the questions I raised earlier. I recommend the paper for publication after two following comments are addressed.

Reply:

We sincerely appreciate your time and effort in reviewing our manuscript. Your constructive feedback has significantly improved the quality of our paper.

(1) One technical detail that I would still like to see is a closer comparison of the ferroelectric and polarization loops. Specifically, I recommend that the authors add an electric field axis in any of the I-V plots in Figure 2. That will bring the comparison closer to Figure 1f.

Thank you for your suggestion. The comparison of the I-V plot and the P-E curve shows that they are well-consistent. From the P-E curve, the coercive field is in the range of $-400 \sim -300$ kV/cm and $400 \sim 550$ kV/cm for coercive fields E_{c-} and E_{c+} , respectively. From I-V curves, the opening of hysteresis window at ~ 500 kV/cm can reflect the coercive field of ferroelectric materials.

Figure R1. Comparison of the I-V curve and the P-E curve.

Following your suggestion, we have added top x-axis for electric fields in Figure 2d,e and

Supplementary Fig. 20a. Also, Figure R1 is added as Supplementary Fig. 2.

(Revised Fig. 2d,e in Page 26)

Supplementary Fig. 2 | Comparison of the I - V curve and the P - E curve regarding electric fields.⁶⁷

(Added Supplementary Fig. 2 in Page 23)

Supplementary Fig. 2 shows that the required electric field for resistance switching is comparable to that for polarization switching. Fig. 2f shows the voltage-sweeping

(Added discussion in Page 5)

(Revised Supplementary Fig. 20a in Page 43)

(2) Are ferroelectric hysteresis loops observed after one or several cycles of resistive switching?

Yes. The ferroelectric hysteresis loops are reproducible with a negligible change after many cycles of resistive switching.